# On the Universal Near Optimality of Hedge in Combinatorial Settings

**Zhiyuan Fan**[*]
MIT
fanzy@mit.edu

**Arnab Maiti**[*]
University of Washington
arnabm2@uw.edu

**Kevin Jamieson**
University of Washington
jamieson@cs.washington.edu

**Lillian J. Ratliff**
University of Washington
ratliffl@uw.edu

**Gabriele Farina**
MIT
gfarina@mit.edu

## Abstract

In this paper, we study the classical HEDGE algorithm in combinatorial settings. In each round, the learner selects a vector $\boldsymbol{x}_t$ from a set $\mathcal{X} \subseteq \{0,1\}^d$, observes a full loss vector $\boldsymbol{y}_t \in \mathbb{R}^d$, and incurs a loss $\langle \boldsymbol{x}_t, \boldsymbol{y}_t \rangle \in [-1,1]$. This setting captures several important problems, including extensive-form games, resource allocation, $m$-sets, online multitask learning, and shortest-path problems on directed acyclic graphs (DAGs). It is well known that HEDGE achieves a regret of $\mathcal{O}\big(\sqrt{T \log |\mathcal{X}|}\big)$ after $T$ rounds of interaction. In this paper, we ask whether HEDGE is optimal across *all* combinatorial settings. To that end, we show that for any $\mathcal{X} \subseteq \{0,1\}^d$, HEDGE is near-optimal—specifically, up to a $\sqrt{\log d}$ factor—by establishing a lower bound of $\Omega\big(\sqrt{T \log(|\mathcal{X}|)/\log d}\big)$ that holds for any algorithm. We then identify a natural class of combinatorial sets—namely, $m$-sets with $\log d \le m \le \sqrt{d}$—for which this lower bound is tight, and for which HEDGE is provably suboptimal by a factor of exactly $\sqrt{\log d}$. At the same time, we show that HEDGE is optimal for online multitask learning, a generalization of the classical $K$-experts problem. Finally, we leverage the near-optimality of HEDGE to establish the existence of a near-optimal regularizer for online shortest-path problems in DAGs—a setting that subsumes a broad range of combinatorial domains. Specifically, we show that the classical Online Mirror Descent (OMD) algorithm, when instantiated with the dilated entropy regularizer, is iterate-equivalent to HEDGE, and therefore inherits its near-optimal regret guarantees for DAGs.

## 1 Introduction

Prediction with expert advice is a central problem in online learning [31, 13, 11, 14, 2]. In this problem, a learner selects a probability distribution over a set of experts $\{1, 2, \ldots, K\}$ in each round. After making the choice, the learner observes the losses of all experts, which may be assigned adversarially within $[-1, 1]$. The goal is to minimize the *cumulative regret*, defined as the difference between the learner's expected total loss and the loss of the best expert in hindsight after $T$ rounds. A simple and widely used algorithm for this setting is HEDGE, introduced by Freund and Schapire [24], which guarantees a regret bound of $\mathcal{O}(\sqrt{T \log K})$.

Since its introduction, the HEDGE algorithm has been extended to a variety of settings, including adversarial bandits [3], continuous action spaces [30], stochastic regimes [34], discounted losses

---

[*]Equal contribution

39th Conference on Neural Information Processing Systems (NeurIPS 2025).

[23], and adaptive learning rates [18]. An important special case is *combinatorial settings*, where the learner selects a vector $x_t$ from a combinatorial set $\mathcal{X} \subseteq \{0,1\}^d$ in each round, observes a loss vector $\boldsymbol{y}_t$, and incurs a loss of $\langle \boldsymbol{x}_t, \boldsymbol{y}_t \rangle \in [-1,1]$. The objective remains to minimize regret against the best fixed vector $\boldsymbol{x}^* \in \mathcal{X}$ in hindsight.

The combinatorial setting captures a wide variety of problems, including extensive-form games, resource allocation games (*e.g.*, Colonel Blotto problems), online multitask learning problem, $\{0,1\}^d$ hypercube, perfect matchings, spanning trees, cut sets, $m$-sets, and online shortest paths in directed acyclic graphs (DAGs). These problems have wide-ranging applications. For instance, extensive-form games provide a foundational framework for modeling sequential games with imperfect information and have been used to build human-level and even superhuman-level AI agents for real-world games [33, 7–9]. Online shortest path problems in DAGs arise naturally in applications like network routing [4, 17]. Resource allocation games have been widely studied in the context of military strategy, political campaigns, sports, and advertising [6, 1].

Given their broad relevance, these combinatorial problems have been extensively studied through the lens of online learning [44, 28, 29, 12, 16, 36, 45]. In the full-information setting, where the learner observes the entire loss vector, an important class of extensive-form games admits optimal algorithms that are efficiently implementable, with HEDGE shown to satisfy both properties in this context [5, 19]. Another example is the work of Takimoto and Warmuth [44], which provided an efficient implementation of a variant of HEDGE for online shortest path problems in DAGs. In the bandit setting, refined variants of HEDGE achieve minimax-optimal regret [12, 10], though a recent work shows they can still be significantly suboptimal for certain combinatorial families [32].

In this paper, we return to the full-information setting and revisit a natural approach for addressing combinatorial problems: treating each element $\boldsymbol{x} \in \mathcal{X}$ as an expert and directly applying the HEDGE algorithm. This yields a regret bound of $\mathcal{O}\big(\sqrt{T \log |\mathcal{X}|}\big)$. While this naive approach is often computationally intractable due to the size of $\mathcal{X}$, Farina et al. [20] (building on prior ideas by Takimoto and Warmuth [44]) recently showed that HEDGE and its optimistic variants can be implemented efficiently in a variety of important combinatorial settings that admit an efficient kernel. Given HEDGE's fundamental advantages—both its simplicity and its broad applicability—we are motivated to ask the following natural question:

*Does* HEDGE *achieve optimal regret guarantees for every combinatorial set* $\mathcal{X} \subseteq \{0,1\}^d$?

## 1.1 Our Contributions

In this paper, we address the above question by establishing the following results:

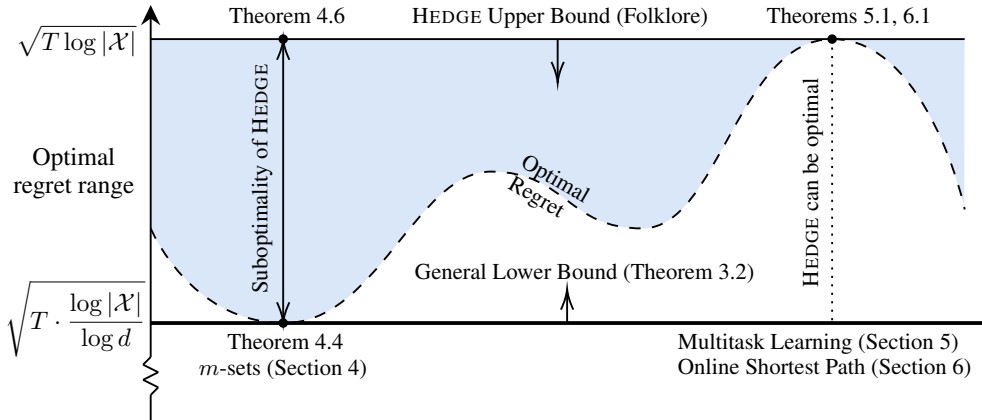

Figure 1: An overview of our results. The $x$-axis indexes different combinatorial decision sets $\mathcal{X} \subseteq \{0,1\}^d$, and the $y$-axis shows the optimal regret over $T$ rounds. We show that HEDGE is near-optimal for all $\mathcal{X}$, up to a $\sqrt{\log d}$ factor. For $m$-sets, HEDGE is provably suboptimal by a factor of $\sqrt{\log d}$, whereas in structured settings such as online multitask learning and important families of DAGs, it is in fact optimal.

- **Universal Near-Optimality:** In Theorem 3.2, we show that HEDGE is universally near optimal for combinatorial games by proving that the regret lower bound for any algorithm on any combinatorial set $\mathcal{X} \subseteq \{0,1\}^d$ is $\Omega\big(\max\{\sqrt{T\log|\mathcal{X}|/\log d}, \sqrt{T}\}\big)$.
- **Suboptimality in Specific Cases:** In Theorems 4.4 and 4.6, we show that the lower bound from Theorem 3.2 is tight for a natural class of combinatorial sets $\mathcal{X}$, and that HEDGE is provably suboptimal for these sets. Specifically, for $m$-sets with $\log d \leq m \leq \sqrt{d}$, we prove that HEDGE necessarily incurs a regret of $\Omega\big(\sqrt{T\log|\mathcal{X}|}\big)$, while we design an Online Mirror Descent (OMD) algorithm that achieves an optimal regret of $\mathcal{O}\big(\sqrt{T\log|\mathcal{X}|/\log d}\big)$.
- **Optimality in Specific Cases:** In Theorem 5.1, we show that HEDGE is optimal for a specific class of combinatorial sets $\mathcal{X}$. In particular, for online multitask learning—which generalizes the classical $K$-experts problem—we prove that any algorithm must incur a regret of $\Omega\big(\sqrt{T\log|\mathcal{X}|}\big)$.

Beyond these results, we further investigate the optimality of HEDGE and its connection to regularization-based algorithms in the structured setting of directed acyclic graphs (DAGs), which capture a broad class of combinatorial problems—including extensive-form games, resource allocation game, $m$-sets, multitask learning, and the $\{0,1\}^d$ hypercube:

- In Theorem 6.1, we show that HEDGE is minimax optimal when $\mathcal{X}$ corresponds to the set of paths from source to sink in a DAG.
- In Section 6.1, we show that OMD with the dilated entropy regularizer achieves a minimax-optimal regret of $\mathcal{O}\big(\sqrt{T\log|\mathcal{X}|}\big)$ for DAGs.
- In Theorem 6.5, we show that OMD with the dilated entropy regularizer is iterate-equivalent to HEDGE on DAGs—thereby inheriting the regret guarantees of HEDGE and demonstrating that HEDGE can be implemented efficiently on DAGs via OMD.

## 1.2 Related Works

**Prediction with Expert Advice.** One of the earliest works on online prediction was by Littlestone and Warmuth [31], who introduced the Weighted Majority algorithm. Cesa-Bianchi et al. [13] extended this line of work by studying the setting where experts' predictions lie in the interval $[0,1]$, while the outcomes are binary. Subsequently, Freund and Schapire [24] addressed the more general setting where both predictions and outcomes lie in $[0,1]$. They proposed the HEDGE algorithm and established a regret bound of $\mathcal{O}(\sqrt{T\log K})$, where $T$ is the time horizon and $K$ is the number of experts.

This foundational result gave rise to several important subsequent works. Erven et al. [18] introduced AdaHedge, a variant of HEDGE with adaptive learning rates that achieves a regret of roughly $\sqrt{L_T^* \log K}$, where $L_T^*$ is the cumulative loss of the best expert. Krichene et al. [30] studied a continuous version of HEDGE for online optimization over compact convex sets $S \subset \mathbb{R}^d$. In the stochastic setting, Mourtada and Gaïffas [34] analyzed HEDGE with decreasing learning rates and obtained a regret bound of $\mathcal{O}(\log N/\Delta)$, where $\Delta$ denotes the sub-optimality gap. For the bandit feedback setting, Auer et al. [3] developed EXP3, a bandit-feedback variant of HEDGE that achieves a regret of $\mathcal{O}(\sqrt{KT})$.

**Combinatorial Settings.** Online learning in combinatorial games has recently received considerable attention. Farina et al. [20] showed that HEDGE and its optimistic variants [15, 37] can be implemented efficiently when the combinatorial game admits an efficient kernel. Examples of such problems include extensive-form games, resource allocation games, $m$-sets, and, more generally, online shortest path problems in directed acyclic graphs (DAGs). Hoda et al. [27] introduced the *dilated entropy* regularizer for extensive-form games and analyzed its properties (cf. also Farina et al. [21]). Building on this, Bai et al. [5] demonstrated that Online Mirror Descent (OMD) with the a specific variant of the dilated entropy regularizer is iterate-equivalent to HEDGE in extensive-form games. Fan et al. [19] subsequently provided the first OMD-based regret analysis for this regularizer, matching the known bounds for HEDGE. Their analysis introduced a new norm, called the *treeplex norm*, to facilitate the regret bounds.

In the bandit feedback setting, Cesa-Bianchi and Lugosi [12] analyzed online learning over specific combinatorial sets $\mathcal{X} \subseteq \{0,1\}^d$, and proposed a variant of HEDGE called COMBAND, achieving an

expected regret bound of $\mathcal{O}\big(\sqrt{dT\log|\mathcal{X}|}\big)$ for those sets. Bubeck et al. [10] subsequently extended this result to *any* combinatorial set $\mathcal{X} \subseteq \{0,1\}^d$, using a variant of HEDGE known as EXP2 with John's exploration. Recently, Maiti et al. [32] showed that this bound is sub-optimal by a factor of $d^{1/4}$ for a specific family of directed acyclic graphs, thereby demonstrating that these variants of HEDGE can, in fact, be substantially sub-optimal.

While the above prior works—as well as our own—focus on loss vectors $\boldsymbol{y}_t$ such that $\langle \boldsymbol{x}, \boldsymbol{y}_t \rangle \in [-1, 1]$ for all $\boldsymbol{x} \in \mathcal{X}$, other works [44, 29, 36] consider coordinate-wise bounded losses, where each $\boldsymbol{y}_t[i] \in [0,1]$ for all $i \in [d]$. Under coordinate-wise bounded losses, Takimoto and Warmuth [44] were the first to implement a variant of HEDGE efficiently on DAGs by leveraging the additivity of losses across the edges of a path. Koolen et al. [29] subsequently analyzed the COMPONENT HEDGE algorithm over various combinatorial sets, including $m$-sets and DAGs. Rahmanian and Warmuth [36] further extended COMPONENT HEDGE to the $k$-multipaths problem.

**Rademacher Complexity.** Orabona and Pál [35] provided a non-asymptotic lower bound of $\Omega(\sqrt{T\log N})$ for the experts problem by analyzing the supremum of a sum of Rademacher random variables. A series of works [39, 38, 22] extended this analysis to more general online learning problems—including combinatorial settings—by characterizing regret in terms of *sequential Rademacher complexity*. Srebro et al. [42] showed that there always exists an instance of Follow-the-Regularized-Leader (FTRL) that is nearly optimal for online linear optimization. This was recently strengthened by Gatmiry et al. [25], who showed that an optimal FTRL instance always exists for online linear optimization. However, the construction of an optimal regularizer for FTRL may incur significant computational overhead.

## 2   Preliminaries

In this paper, we study the repeated online decision-making problem in a combinatorial setting. Denote by $d$ the dimension of the problem. The agent is given a set of discrete actions $\mathcal{X} \subseteq \{0,1\}^d$. At each round $t = 1, 2, \ldots$, the agent selects an action $\boldsymbol{x}_t$ from the decision set $\mathcal{X}$. The environment simultaneously chooses a loss vector $\boldsymbol{y}_t \in \mathcal{Y}$, potentially adversarially based on the interaction history $\mathcal{F}_t := \{(\boldsymbol{x}_\tau, \boldsymbol{y}_\tau)\}_{\tau=1}^{t-1}$. The agent then incurs a loss of $\langle \boldsymbol{x}_t, \boldsymbol{y}_t \rangle$ and observes the loss vector $\boldsymbol{y}_t$. The goal of the agent is to minimize the total loss over $T$ rounds, or equivalently, to minimize the cumulative regret:

$$\text{Regret}(T) := \sum_{t=1}^{T} \langle \boldsymbol{x}_t, \boldsymbol{y}_t \rangle - \min_{\boldsymbol{x}_* \in \mathcal{X}} \sum_{t=1}^{T} \langle \boldsymbol{x}_*, \boldsymbol{y}_t \rangle.$$

We focus on the combinatorial game setting, in which the loss vector is restricted so that the loss incurred at each round is bounded within $[-1, 1]$. In this case, the loss vector set $\mathcal{Y}$ is the polar set of the convex hull $\text{co}(\mathcal{X})$. We note that this assumption covers various settings in the literature, including extensive-form games [20, 5]. Formally, we make the following assumption:

**Assumption 2.1.** The loss vector set is defined as $\mathcal{Y} := \big\{ \boldsymbol{y} \in \mathbb{R}^d : \max_{\boldsymbol{x} \in \mathcal{X}} |\langle \boldsymbol{x}, \boldsymbol{y} \rangle| \leq 1 \big\}$.

**The Hedge Algorithm**   A classical approach for solving this problem is the HEDGE algorithm [24], also known as Multiplicative Weight Updates (MWU). In this algorithm, the agent chooses actions in a randomized manner based on the past performance of the actions. Specifically, let $\eta > 0$ be the learning rate, the probability of choosing an action $\boldsymbol{x}$ in round $t$ is proportional to

$$\mathbb{P}_t(\boldsymbol{x}) \propto w_t(\boldsymbol{x}) := \exp\left( -\eta \sum_{\tau=1}^{t-1} \langle \boldsymbol{x}, \boldsymbol{y}_\tau \rangle \right), \qquad \forall \boldsymbol{x} \in \mathcal{X}.$$

A classical result shows under learning rate $\eta := \sqrt{\log|\mathcal{X}|/T}$, this algorithm has a regret upper bound of $\mathcal{O}\big(\sqrt{T\log|\mathcal{X}|}\big)$ in the combinatorial game setting (i.e., under Assumption 2.1). Furthermore, the algorithm requires $\mathcal{O}(d|\mathcal{X}|)$ time per round to compute the updates. This may be exponential in $d$ when only a succinct representation of the decision set $\mathcal{X}$ is provided. It is known that when the *kernel* of the decision set $\mathcal{X}$ can be computed efficiently, it is possible to simulate the MWU algorithm in polynomial time using the KERNELIZED MWU algorithm [20].

**Proximal Methods** We recall the foundational concepts and notations commonly used in proximal optimization methods. Let $\mathcal{X}$ denote the decision set. The proximal methods is built upon on some regularizer $\varphi : \mathrm{co}(\mathcal{X}) \to \mathbb{R}$, which is required to be $\mu$-strongly convex with respect to a chosen norm $\|\cdot\|$ over $\mathrm{co}(\mathcal{X})$. Such a function naturally gives rise to a generalized measure of divergence known as the *Bregman divergence*, defined for any vectors $\boldsymbol{x}', \boldsymbol{x} \in \mathcal{X}$ as:

$$\mathcal{D}_\varphi(\boldsymbol{x}' \,\|\, \boldsymbol{x}) := \varphi(\boldsymbol{x}') - \varphi(\boldsymbol{x}) - \langle \nabla\varphi(\boldsymbol{x}), \boldsymbol{x}' - \boldsymbol{x} \rangle.$$

Among proximal methods, a general approach to solving the online decision problem is the Online Mirror Descent (OMD) algorithm [15]. Let $\eta > 0$ be the learning rate, and let $\varphi : \mathrm{co}(\mathcal{X}) \to \mathbb{R}$ be a strongly convex regularizer. The algorithm maintains a policy in each round $t$. In the first round, the policy is set so the regularizer is unique minimizer: $\widetilde{\boldsymbol{x}}_1 \leftarrow \mathrm{argmin}_{\boldsymbol{x} \in \mathrm{co}(\mathcal{X})} \varphi(\boldsymbol{x})$. For each round $t = 2, 3, \ldots$, the agent takes proximal step:

$$\widetilde{\boldsymbol{x}}_t \leftarrow \underset{\boldsymbol{x} \in \mathrm{co}(\mathcal{X})}{\mathrm{argmin}} \left\{ \eta \langle \boldsymbol{y}_{t-1}, \boldsymbol{x} \rangle + \mathcal{D}_\varphi(\boldsymbol{x} \,\|\, \widetilde{\boldsymbol{x}}_{t-1}) \right\}$$

Then, the agent draws and plays an action $\boldsymbol{x}_t \in \mathcal{X}$ by matching its expectation to the proposed policy, i.e., $\mathbb{E}_t[\boldsymbol{x}_t] = \widetilde{\boldsymbol{x}}_t$. It is known that this algorithm achieves the following regret upper bound:

**Theorem 2.2** (Regret Bound for OMD, [37, 43]). Let $\|\cdot\|$ and $\|\cdot\|_*$ be a pair of primal-dual norm defined on $\mathbb{R}^d$. Let $\varphi$ be a DGF that is $\mu$-strongly convex on $\|\cdot\|$. Denote $\boldsymbol{y}_t$ as the reward gradient received in episode $t$. The cumulative regret of running OMD with DGF $\varphi$ and learning rate $\eta > 0$ can be upper bounded by

$$\widetilde{\mathrm{Regret}}(T) := \sum_{t=1}^{T} \langle \widetilde{\boldsymbol{x}}_t, \boldsymbol{y}_t \rangle - \min_{\boldsymbol{x}_* \in \mathcal{X}} \sum_{t=1}^{T} \langle \boldsymbol{x}_*, \boldsymbol{y}_t \rangle \leq \frac{1}{\eta} \mathcal{D}_\varphi(\boldsymbol{x}_* \,\|\, \widetilde{\boldsymbol{x}}_1) + \frac{\eta}{2\mu} \sum_{t=1}^{T} \|\boldsymbol{y}_t\|_*^2.$$

**General Notation** We use lowercase boldface letters, such as $\boldsymbol{x}$, to denote vectors. The notation $|\boldsymbol{x}|$ denotes the element-wise absolute value. For an index set $\mathcal{C}$, let $\boldsymbol{x}[\mathcal{C}] \in \mathbb{R}^{\mathcal{C}}$ denote the subvector of $\boldsymbol{x}$ restricted to entries indexed by $\mathcal{C}$, and let $|\mathcal{C}|$ denote the cardinality of the set. We write $[\![k]\!] := \{1, 2, \ldots, k\}$ and use $\varnothing$ to denote the empty set. The probability simplex over a finite set $\mathcal{C}$ is denoted by $\mathcal{P}(\mathcal{C})$. We use $\log x$ to denote the logarithm of $x$ in base 2 and $\ln x$ to denote the natural logarithm of $x$. For non-negative sequences $\{a_n\}$ and $\{b_n\}$, we write $a_n \leq \mathcal{O}(b_n)$ or equivalently $b_n \geq \Omega(a_n)$ to indicate the existence of a global constant $C > 0$ such that $a_n \leq Cb_n$ for all $n > 0$. Similarly, we write $a_n = \Theta(b_n)$ to indicate the existence of global constants $C_1, C_2 > 0$ such that $C_1 b_n \leq a_n \leq C_2 b_n$ for all $n > 0$. Lastly, we denote by $\mathrm{co}(\mathcal{X})$ the convex hull of a set $\mathcal{X}$.

## 3 Universal Near Optimality of Hedge

In this section, we show that for any given combinatorial decision set $\mathcal{X} \subseteq \{0,1\}^d$, the HEDGE algorithm achieves a near optimal regret bound in the combinatorial game setting. We begin by stating the following classical result.

**Lemma 3.1** (Sauer–Shelah Lemma [40, 41]). Let $\mathcal{C}$ be a family of sets whose union has $n$ elements. A set $S$ is said to be *shattered* by $\mathcal{C}$ if every subset of $S$ can be obtained as the intersection $S \cap C$ for some set $C \in \mathcal{C}$. $\mathcal{C}$ shatters a set of size $k$, if the number of sets in the family satisfies

$$|\mathcal{C}| > \sum_{i=0}^{k-1} \binom{n}{i}.$$

By the Sauer–Shelah Lemma, there exists an index set $\mathcal{I} \subseteq [\![d]\!]$ of size $\Omega(\log|\mathcal{X}|/\log d)$ such that the restriction of $\mathcal{X}$ to the coordinates in $\mathcal{I}$ equals the full hypercube $\{0,1\}^{\mathcal{I}}$. Consequently, any hard instance with loss vectors supported only on coordinates in $\mathcal{I}$ is at least as hard as the corresponding instance of the combinatorial game over the hypercube $\mathcal{X}' := \{0,1\}^{\mathcal{I}}$. As any algorithm suffers a regret lower bound of $\Omega\big(\sqrt{T\,|\mathcal{I}|}\big)$ in the combinatorial game over the hypercube. This yields a regret lower bound of $\Omega\big(\sqrt{T\log|\mathcal{X}|/\log d}\big)$.

We formally state our main result in the following theorem. We defer the proof to Appendix A.

**Theorem 3.2.** Let $\mathcal{X} \subseteq \{0,1\}^d$ be a decision set and let $\mathcal{Y}$ be the corresponding loss vector set that satisfies Assumption 2.1. For any $T \geq 1$ and any Algorithm ALG, there exists a sequence of loss vectors $\boldsymbol{y}_1, \boldsymbol{y}_2, \ldots, \boldsymbol{y}_T \in \mathcal{Y}$ such that the algorithm incurs an expected regret of at least

$$\mathbb{E}[\text{Regret}(T)] = \mathbb{E}\left[\sum_{t=1}^{T} \langle \boldsymbol{x}_t, \boldsymbol{y}_t \rangle - \min_{\boldsymbol{x}_* \in \mathcal{X}} \sum_{t=1}^{T} \langle \boldsymbol{x}_*, \boldsymbol{y}_t \rangle\right] \geq \Omega\left(\max\left\{\sqrt{T \cdot \frac{\log|\mathcal{X}|}{\log d}}, \sqrt{T}\right\}\right).$$

The expectation is taken over any potential randomness of the algorithm.

## 4 The Sub-Optimality of Hedge on $m$-Sets

In this section, we consider a specific decision set $\mathcal{X}$—namely, the family of $m$-sets—to illustrate the suboptimality of the HEDGE algorithm. For an integer $m \in [\![d/2]\!]$, the $m$-sets problem corresponds to the decision set $\mathcal{X} := \{\boldsymbol{x} \in \{0,1\}^d : \sum_{i=1}^{d} \boldsymbol{x}[i] = m\}$. We show that OMD, with a suitable regularizer, matches the regret lower bound from Theorem 3.2 when $\log d \leq m \leq d/2$, whereas HEDGE suffers a regret lower bound of $\Omega(\sqrt{T \log|\mathcal{X}|})$, establishing a suboptimality gap of $\sqrt{\log d}$.

### 4.1 The Regret Upper Bound of $m$-Sets

We begin by presenting our OMD algorithm for $m$-sets, for any $m \in [\![d/2]\!]$. Previous work [42] shows that there always exists a regularizer that enables the OMD algorithm to achieve a near-optimal regret bound. However, the regularizer is not constructive, and the corresponding regret bound is implicit. Instead, we need to construct a regularizer suitable for the decision set so that the regret bound in Theorem 2.2 is minimized. Specifically, we analyze the OMD algorithm with the following regularizer:

$$\varphi(\boldsymbol{x}) := \sum_{i=1}^{d}\left(\boldsymbol{x}[i]^2 + \frac{1}{m}\boldsymbol{x}[i]\ln\boldsymbol{x}[i]\right). \tag{4.1}$$

According to Theorem 2.2, it suffices to pick a pair of primal-dual norm $\|\cdot\|$ and $\|\cdot\|_*$ and analyze the strong convexity of $\varphi$ and also the vector norm $\|\boldsymbol{y}_t\|_*$. We define a pair of dual-primal norms:

$$\|\boldsymbol{z}\|_* := \max_{\boldsymbol{x} \in \mathcal{X}} |\langle \boldsymbol{x}, \boldsymbol{z} \rangle|, \qquad \|\boldsymbol{z}\| := \max_{\|\boldsymbol{y}\|_* \leq 1} \langle \boldsymbol{y}, \boldsymbol{z} \rangle.$$

First, we state the following proposition that establishes an upper bound on the primal norm $\|\boldsymbol{z}\|$. The proof is done by direct calculation, and we defer the details to Appendix B.

**Proposition 4.1.** For any vector $\boldsymbol{z} \in \mathbb{R}^d$, the primal norm $\|\cdot\|$ is upper bounded by the $\ell_1$ and $\ell_\infty$ norms together, namely,

$$\|\boldsymbol{z}\| \leq 3\|\boldsymbol{z}\|_\infty + \frac{1}{m}\|\boldsymbol{z}\|_1.$$

By applying the above proposition, we establish the strong convexity of the regularizer $\varphi$ with respect to the primal norm. We defer the proof to Appendix B.

**Lemma 4.2.** The function $\varphi$ is $1/9$-strongly convex with respect to the primal norm $\|\cdot\|$.

The following lemma bounds the Bregman divergence under the regularizer $\varphi$.

**Lemma 4.3.** We have $\mathcal{D}_\varphi(\boldsymbol{x}_* \,\|\, \widetilde{\boldsymbol{x}}_1) \leq m + \ln(d/m)$, for any vector $\boldsymbol{x}_* \in \mathcal{X}$.

We defer the proof to Appendix B. Using the above results, we are able to establish the regret upper bound for running OMD with the regularizer defined in (4.1).

**Theorem 4.4.** Let $1 \leq m \leq d/2$. With the choice $\eta := \sqrt{2(m + \ln(d/m))/(9T)}$, the expected regret of running OMD with the regularizer in (4.1) over $m$-sets is upper bounded by

$$\mathbb{E}[\text{Regret}(T)] \leq \mathcal{O}\left(\sqrt{Tm + T\log(d/m)}\right).$$

*Proof.* According to the definition of $\|\cdot\|_*$, we have that $\|\boldsymbol{y}_t\|_* \leq 1$. Combining this with Lemma 4.2 and Lemma 4.3, and applying Theorem 2.2 together with $\mathbb{E}[\boldsymbol{x}_t] = \widetilde{\boldsymbol{x}}_t$, we conclude that

$$\mathbb{E}[\text{Regret}(T)] \leq \frac{1}{\eta}\mathcal{D}_\varphi(\boldsymbol{x}_* \,\|\, \widetilde{\boldsymbol{x}}_1) + \frac{\eta}{2\mu}\sum_{t=1}^T \|\boldsymbol{y}_t\|_*^2$$
$$\leq \frac{1}{\eta}\big(m + \ln(d/m)\big) + \frac{9\eta}{2}\cdot T$$
$$\leq \sqrt{18Tm + 18T\ln(d/m)},$$

where the last inequality is given by the choice $\eta$. $\qquad\square$

We show that this regret upper bound is in fact optimal by establishing a matching lower bound. Specifically, we construct our lower bound using the hard instance from Theorem 3.2 along with the hard instance for the $K$-experts problem (see Lemma F.4). Formally, we show the following:

**Theorem 4.5.** Consider integers $m, d, T$ such that $1 \leq m \leq d/2 \leq \exp(T/3)/2$. For the $m$-sets problem and any Algorithm ALG, there exists a sequence of loss vectors $\boldsymbol{y}_1, \boldsymbol{y}_2, \ldots, \boldsymbol{y}_T \in \mathcal{Y}$ such that the algorithm incurs a expected regret of at least

$$\mathbb{E}[\text{Regret}(T)] = \mathbb{E}\left[\sum_{t=1}^T \langle \boldsymbol{x}_t, \boldsymbol{y}_t \rangle - \min_{\boldsymbol{x}_* \in \mathcal{X}} \sum_{t=1}^T \langle \boldsymbol{x}_*, \boldsymbol{y}_t \rangle\right] \geq \Omega\Big(\sqrt{Tm + T\log(d/m)}\Big).$$

We defer the proof to Appendix B.

## 4.2 The Regret Lower Bound of Hedge

We introduce the following theorem, showing that the HEDGE algorithm is *strictly* sub-optimal on $m$-sets, when $\log d \leq m \leq \sqrt{d}$. The full proof is deferred to Appendix B.

**Theorem 4.6.** Consider integers $m, d, T$ such that $1 \leq m \leq d/2 \leq \exp(T/3)/2$ and $m\log(d/m) \leq T$. For any $\eta > 0$, there exists a there exists a sequence of loss vectors $\boldsymbol{y}_1, \boldsymbol{y}_2, \ldots, \boldsymbol{y}_T \in \mathcal{Y}$ over $m$-sets such that the HEDGE algorithm with learning rate $\eta$ incurs a expected regret of at least

$$\mathbb{E}[\text{Regret}(T)] = \mathbb{E}\left[\sum_{t=1}^T \langle \boldsymbol{x}_t, \boldsymbol{y}_t \rangle - \min_{\boldsymbol{x}_* \in \mathcal{X}} \sum_{t=1}^T \langle \boldsymbol{x}_*, \boldsymbol{y}_t \rangle\right] \geq \Omega\Big(\sqrt{Tm\log(d/m)}\Big).$$

*Proof sketch.* We divide the proof into two cases based on how $\eta$ compares to the base learning rate $\eta_0 := \sqrt{T^{-1}m\ln(d/m)} = \Theta\big(\sqrt{T^{-1}\log|\mathcal{X}|}\big)$.

When the learning rate is small, i.e., $\eta \leq \eta_0$, we construct a hard instance by assigning the same fixed loss vector $\boldsymbol{y}_t$ across all rounds, where $\boldsymbol{y}_t[i] := \mathbb{1}[i \in \llbracket m \rrbracket]/m$ for all $i \in \llbracket d \rrbracket$. In this case, we can show that HEDGE with a small learning rate incurs a constant regret for any round $t \leq t_0 := \Omega\big(\sqrt{Tm\log(d/m)}\big)$. Thus, we establish a regret lower bound of $\Omega\big(\sqrt{Tm\log(d/m)}\big)$.

When the learning rate is large, i.e., $\eta > \eta_0$, we construct a hard instance by setting $\boldsymbol{y}_t[i] = 0$ for all coordinates $i \geq 2$ across all rounds. For the first coordinate, the loss $\boldsymbol{y}_t[1]$ is assigned in two phases, based on the threshold $t_0 := \ln(d/m)/\eta$ (assuming $t_0 \in \mathbb{N}$ for simplicity). In **Phase 1** (rounds $t \leq t_0$), we set $\boldsymbol{y}_t[1] = -1$. In **Phase 2** (rounds $t > t_0$), the value of $\boldsymbol{y}_t[1]$ alternates: it is 1 if $t - t_0$ is odd, and $-1$ if $t - t_0$ is even.

In this case, we can show that HEDGE with a large learning rate incurs a regret of $\Omega\big(\min\{\eta, 1\}\big)$ for every two rounds after $t > t_0$. Thus, we establish a regret lower bound via

$$\mathbb{E}[\text{Regret}(T)] \geq (T - t_0) \cdot \Omega\big(\min\{\eta, 1\}\big) \geq \Omega\Big(\sqrt{Tm\log(d/m)}\Big).$$

In general, we conclude that HEDGE has a regret lower bound of $\Omega\big(\sqrt{Tm\log(d/m)}\big)$ on the $m$-sets for any learning rate $\eta > 0$. $\qquad\square$

# 5 The Optimality of Hedge for Online Multitask Learning

In the previous section, we showed that the HEDGE algorithm can be strictly suboptimal for certain combinatorial decision sets, such as $m$-sets. This naturally leads to the question: are there combinatorial settings where HEDGE remains optimal? In this section, we answer this in the affirmative by analyzing the Online Multitask Learning problem—a setting that generalizes the classical $K$-experts problem and has been studied in the bandit learning literature as the Multi-Task Bandit problem. In this problem, the learner is presented with $m \geq 1$ separate expert problems, where the $i$-th problem involves $d_i \geq 2$ experts. In each round, the learner selects one expert from each problem and incurs a loss that is the sum of the losses associated with the chosen experts. The goal is to minimize regret with respect to the best expert in each problem in hindsight.

We parameterize the online multitask learning problem as follows: Let $d_{1:i} := \sum_{j=1}^{i} d_j$ be the total number of experts in the first $i$ expert problems, with $d_{1:0} := 0$. The decision set $\mathcal{X}$ is of dimension $d = d_{1:m}$ given by

$$\mathcal{X} = \left\{ \boldsymbol{x} \in \{0,1\}^d : \sum_{j=d_{1:i-1}+1}^{d_{1:i}} \boldsymbol{x}[j] = 1, \; \forall i \in [\![m]\!] \right\}.$$

Recall that the adversary is restricted to choose $\boldsymbol{y}_t$ such that $\langle \boldsymbol{x}, \boldsymbol{y}_t \rangle \in [-1, 1]$ for all $\boldsymbol{x} \in \mathcal{X}$ in each round $t$ following from Assumption 2.1. In this case, HEDGE has a regret upper bound of $\mathcal{O}\left(\sqrt{T \log |\mathcal{X}|}\right)$. In the following theorem we show that HEDGE is optimal for the online multitask Learning problem. We construct a hard instance by using the hard instance for the $i$-th expert problem over $\frac{\log d_i}{\sum_{j=1}^{m} \log d_j} \cdot T$ rounds. The full proof is deferred to Appendix C.

**Theorem 5.1.** Consider any instance of the online multitask learning problem on $\mathcal{X}$ of dimension $d \geq 2$. Let the corresponding loss vector set $\mathcal{Y}$ satisfy Assumption 2.1. Consider an integer $T \geq 3 \log |\mathcal{X}|$. Then for any Algorithm ALG, there exists a sequence of loss vectors $\boldsymbol{y}_1, \boldsymbol{y}_2, \ldots, \boldsymbol{y}_T \in \mathcal{Y}$ such that the algorithm incurs a regret of at least

$$\mathbb{E}[\text{Regret}(T)] = \mathbb{E}\left[ \sum_{t=1}^{T} \langle \boldsymbol{x}_t, \boldsymbol{y}_t \rangle - \min_{\boldsymbol{x}_* \in \mathcal{X}} \sum_{t=1}^{T} \langle \boldsymbol{x}_*, \boldsymbol{y}_t \rangle \right] \geq \Omega\left(\sqrt{T \log |\mathcal{X}|}\right).$$

# 6 Minimax Optimal Regularizers for Directed Acyclic Graphs

We consider the online shortest path problem in the Directed Acyclic Graphs (DAGs). Let $G = (V, E)$ be a DAG with the source vertex $\mathsf{s} \in V$ and the sink $\mathsf{t} \in V$. We assume that every vertex $v \in V$ is reachable from $\mathsf{s}$ and can reach $\mathsf{t}$. Denote by $\mathcal{X} \subseteq \{0,1\}^E$ the set of all $\mathsf{s}$-$\mathsf{t}$ paths of the graph $G$, indexed by the edges in $E$. Each vertex $\boldsymbol{x} \in \mathcal{X}$ encodes a $\mathsf{s}$-$\mathsf{t}$ path in the graph, where $\boldsymbol{x}[e] = 1$ indicates that $e \in E$ appears in the path. The convex hull of $\mathcal{X}$ forms the flow polytope:

$$\text{co}(\mathcal{X}) = \left\{ \boldsymbol{x} \in [0,1]^E : \sum_{e \in \delta^+(\mathsf{s})} \boldsymbol{x}[e] = \sum_{e \in \delta^-(\mathsf{t})} \boldsymbol{x}[e] = 1 \quad \text{and} \quad \sum_{e \in \delta^-(v)} \boldsymbol{x}[e] = \sum_{e \in \delta^+(v)} \boldsymbol{x}[e], \quad \forall v \in V \right\},$$

where $\delta^-(v) = \{(u,v) \in E\}$ and $\delta^+(v) = \{(v,w) \in E\}$ denotes the set of incoming edges and outgoing edges, respectively. We note that in this case, the loss vector $\boldsymbol{y} \in \mathcal{Y} \subseteq \mathbb{R}^E$ is an assignment of the weights such that any $\mathsf{s}$-$\mathsf{t}$ path has a weight between $-1$ and $1$.

In the following theorem, we show that HEDGE is minimax optimal for DAGs. The proof involves a careful construction of a DAG, parameterized by upper bounds on the number of edges and paths. The full proof is deferred to Appendix D.

**Theorem 6.1.** For any integers $d, N, T$ such that $16 \leq 2d \leq N \leq 2^d$ and $3 \log N \leq T$, and for any algorithm ALG, there exists a DAG $G$ with at most $d$ edges and at most $N$ paths from source $\mathsf{s}$ to sink $\mathsf{t}$, and a corresponding sequence of loss vectors $\boldsymbol{y}_1, \boldsymbol{y}_2, \ldots, \boldsymbol{y}_T \in \mathcal{Y}$ such that the algorithm incurs a regret lower bound of

$$\mathbb{E}[\text{Regret}(T)] = \mathbb{E}\left[ \sum_{t=1}^{T} \langle \boldsymbol{x}_t, \boldsymbol{y}_t \rangle - \min_{\boldsymbol{x}_* \in \mathcal{X}} \sum_{t=1}^{T} \langle \boldsymbol{x}_*, \boldsymbol{y}_t \rangle \right] \geq \Omega\left(\sqrt{T \log N}\right).$$

## 6.1 OMD with Dilated Entropy Regularizer

While our main focus has been on the HEDGE algorithm, it is natural to consider its close counterpart, Online Mirror Descent (OMD). With an appropriate distance-generating function, OMD is known to be iterate-equivalent to HEDGE in extensive-form games [5, 19]. Since DAGs can model such games [32], and HEDGE is minimax-optimal on DAGs, this motivates a closer examination of OMD on DAGs. In this section, we analyze OMD with the dilated entropy regularizer on DAGs and show that it also achieves minimax-optimal regret. Formally, the dilated entropy $\psi \colon \mathrm{co}(\mathcal{X}) \to \mathbb{R}$ is defined by

$$\psi(\boldsymbol{x}) := \sum_{e \in E} \boldsymbol{x}[e] \ln \boldsymbol{x}[e] - \sum_{v \in V} \boldsymbol{x}[v] \ln \boldsymbol{x}[v] = \sum_{v \in V \setminus \{\mathsf{t}\} : \boldsymbol{x}[v] > 0} \sum_{e \in \delta^+(v)} \boldsymbol{x}[e] \ln \frac{\boldsymbol{x}[e]}{\boldsymbol{x}[v]},$$

where, by standard convention, $0 \ln 0 := 0$. Here, $\boldsymbol{x}[v] := \sum_{e \in \delta^+(v)} \boldsymbol{x}[e]$ for all $v \neq \mathsf{t}$, and $\boldsymbol{x}[\mathsf{t}] := 1$.

We note that the regularizer on the policy $\widetilde{\boldsymbol{x}} \in \mathrm{co}(\mathcal{X})$ is closely related to the Shannon entropy over the chosen action $\boldsymbol{x} \in \mathcal{X}$. In fact, consider the following procedure for sampling an action $\boldsymbol{x} \sim \mathcal{D}(\widetilde{\boldsymbol{x}}) \in \mathcal{P}(\mathcal{X})$: we start from the active vertex $u \leftarrow \mathsf{s}$. At each step, we first set $\boldsymbol{x}[u] = 1$, then randomly pick an edge $e = (u, v) \in \delta^+(u)$ with probability $\widetilde{\boldsymbol{x}}[e]/\widetilde{\boldsymbol{x}}[u]$, set $\boldsymbol{x}[e] = 1$, and move to $u \leftarrow v$. Following this Markovian sampling procedure, one can see that the drawn vector $\boldsymbol{x}$ is consistent with $\widetilde{\boldsymbol{x}}$, i.e., $\mathbb{E}[\boldsymbol{x}] = \widetilde{\boldsymbol{x}}$. Furthermore, we have the following:

**Lemma 6.2.** For any $\widetilde{\boldsymbol{x}} \in \mathrm{co}(\mathcal{X})$, we have

$$\psi(\widetilde{\boldsymbol{x}}) = -H(\boldsymbol{x}) := \mathbb{E}_{\boldsymbol{x} \sim \mathcal{D}(\widetilde{\boldsymbol{x}})}[\ln \mathbb{P}(\boldsymbol{x})],$$

where $H(\cdot)$ denotes the Shannon entropy of the random variable.

The proof of the above lemma is deferred to Appendix D. We will now show that running OMD with the dilated regularizer enjoys a regret upper bound of $\mathcal{O}\big(\sqrt{T \log |\mathcal{X}|}\big)$, based on the OMD regret bound in Theorem 2.2. From Lemma 6.2, we have that $\psi$ is equivalent to the negative entropy of distribution over $\mathcal{X}$. This indicates $\mathcal{D}_\psi(\boldsymbol{x}_* \parallel \widetilde{\boldsymbol{x}}_1) \leq \ln |\mathcal{X}|$. It remains to pick the norm functions and show the strong convexity. Consider a pair of primal dual norms

$$\|\boldsymbol{z}\|_* := \max_{\boldsymbol{x} \in \mathcal{X}} |\langle \boldsymbol{x}, \boldsymbol{z} \rangle|, \qquad \|\boldsymbol{z}\| := \max_{\|\boldsymbol{y}\|_* \leq 1} \langle \boldsymbol{y}, \boldsymbol{z} \rangle.$$

We note that since $\mathrm{co}(\mathcal{X})$ is the flow polytope, its dual, $\{\boldsymbol{y} : \|\boldsymbol{y}\|_* \leq 1\}$, is closely related to the set of all cuts of the graph. The next lemma shows $\psi$ is strongly convex over primal norm $\|\cdot\|$. We defer the proof to Appendix D.

**Lemma 6.3.** The function $\psi$ is $1/10$-strongly convex with respect to the primal norm $\|\cdot\|$ in $\mathrm{span}(\mathcal{X})$.

From the standard OMD analysis, running OMD under the dilated entropy $\psi$ achieves a regret upper bound of $\mathcal{O}\big(\sqrt{T \log |\mathcal{X}|}\big)$. Hence, OMD with dilated entropy is minimax optimal for DAGs.

## 6.2 Equivalence of Dilated Entropy and HEDGE

As shown earlier, both OMD with the dilated entropy regularizer and HEDGE over the set of paths in a DAG achieve minimax optimal regret. This naturally raises a fundamental question: are these two approaches equivalent? In this section, we answer this question in the affirmative.

Let $G = (V, E)$ be a directed acyclic graph (DAG) with a designated source vertex $\mathsf{s}$ and sink vertex $\mathsf{t}$, and let $\mathcal{X}$ denote the set of all paths from $\mathsf{s}$ to $\mathsf{t}$. If $G$ contains only a single source-to-sink path, the equivalence is immediate. Therefore, we focus on the case where $G$ admits multiple such paths.

We begin by state the following lemma, the proof of which is deferred to Appendix D.

**Lemma 6.4.** The dilated entropy $\psi$ is differentiable and strictly convex on the relative interior $C := \mathrm{relint}(\mathrm{co}(\mathcal{X}))$. Moreover, $\lim_{n \to \infty} \|\nabla_{\boldsymbol{x}} \psi(\boldsymbol{x}_n)\|_2 = \infty$ if $\{\boldsymbol{x}_n\}_n$ is sequence of points in $C$ approaching the boundary of $C$.

Now, we present the following theorem, which demonstrate that OMD is, in fact, iterate-equivalent to HEDGE. The proof proceeds by formulating the KKT conditions and applying Lemma 6.4 to establish the equivalence. The proof is deferred to Appendix D.

**Theorem 6.5.** OMD with dilated entropy is iterate-equivalent to HEDGE over the set of paths $\mathcal{X}$.

## 7 Conclusion and Future works

We investigated the optimality of the classical HEDGE algorithm in combinatorial online learning settings. While HEDGE achieves a regret of $\mathcal{O}\big(\sqrt{T \log |\mathcal{X}|}\big)$, we established that this rate is nearly optimal—up to a $\sqrt{\log d}$ factor—for any set $\mathcal{X} \subseteq \{0,1\}^d$. We further identified a class of $m$-sets for which HEDGE is provably suboptimal and showed that it remains optimal for the multitask learning problem. Finally, we demonstrated that Online Mirror Descent with the dilated entropy regularizer is iterate-equivalent to HEDGE on DAGs, providing a computationally efficient regularization framework for a broad family of combinatorial domains.

Our work opens up several interesting directions for future research. One natural question is whether there exists a family of efficiently constructible regularizers that are near-optimal for the combinatorial sets. We conjecture that negative entropy in a suitably lifted space may serve as such a regularizer. In support of this, we refer the reader to Appendix E, where we show that the conjecture holds for DAGs. Another compelling direction is to explore whether there exist near-optimal variants of the Follow-the-Perturbed-Leader algorithm for the combinatorial sets. Since perturbations are often considered more implementation-friendly, exploring near-optimal variants of the Follow-the-Perturbed-Leader algorithm could yield both theoretical and practical advances. Finally, we ask whether there are variants of HEDGE that achieve near-optimal regret for arbitrary finite subsets of $\mathbb{R}^d$.

## Acknowledgments

Ratliff is funded in part by ONR YIP N000142012571, and NSF awards 1844729 and 2312775. Jamieson is funded in part by NSF Award CAREER 2141511 and Microsoft Grant for Customer Experience Innovation. Farina is funded in part by NSF Award CCF-2443068, ONR grant N00014-25-1-2296, and an AI2050 Early Career Fellowship.

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

## A  Deferred Proofs from Section 3

**Theorem 3.2.** Let $\mathcal{X} \subseteq \{0,1\}^d$ be a decision set and let $\mathcal{Y}$ be the corresponding loss vector set that satisfies Assumption 2.1. For any $T \geq 1$ and any Algorithm ALG, there exists a sequence of loss vectors $\boldsymbol{y}_1, \boldsymbol{y}_2, \ldots, \boldsymbol{y}_T \in \mathcal{Y}$ such that the algorithm incurs an expected regret of at least

$$\mathbb{E}[\mathrm{Regret}(T)] = \mathbb{E}\left[\sum_{t=1}^T \langle \boldsymbol{x}_t, \boldsymbol{y}_t \rangle - \min_{\boldsymbol{x}_* \in \mathcal{X}} \sum_{t=1}^T \langle \boldsymbol{x}_*, \boldsymbol{y}_t \rangle\right] \geq \Omega\left(\max\left\{\sqrt{T \cdot \frac{\log|\mathcal{X}|}{\log d}}, \sqrt{T}\right\}\right).$$

The expectation is taken over any potential randomness of the algorithm.

*Proof.* If $|\mathcal{X}| = 1$, the result holds trivially. Therefore, we assume $|\mathcal{X}| \geq 2$. We begin by considering a deterministic algorithm ALG and establish a regret lower bound for it. The result is then extended to randomized algorithms via Yao's lemma.

Let $k := \max\left\{\left\lfloor \frac{\log|\mathcal{X}|}{\log(2ed)} \right\rfloor, 1\right\}$. We will show that $|\mathcal{X}| > \sum_{i=0}^{k-1} \binom{d}{i}$ in general. This is clear when $k = 1$, as we have $|\mathcal{X}| \geq 2 > 1 = \sum_{i=0}^{k-1} \binom{d}{i}$. In the case where $k \in [\![2, d]\!]$, we have that

$$|\mathcal{X}| \geq (2ed)^k > k \cdot \left(\frac{2ed}{k}\right)^k \geq \sum_{i=1}^k \binom{2d}{k} > \sum_{i=1}^k \binom{2d}{i} > \sum_{i=0}^{k-1} \binom{2d}{i} > \sum_{i=0}^{k-1} \binom{d}{i}.$$

For every vector $\boldsymbol{x} \in \mathcal{X}$, we construct a corresponding indicator set $C_{\boldsymbol{x}} := \{i \in [\![d]\!] : \boldsymbol{x}[i] = 1\}$. Denote by $\mathcal{C} := \{C_{\boldsymbol{x}} : \boldsymbol{x} \in \mathcal{X}\}$ be the collection of such sets. According to Lemma 3.1, there exists a set $\mathcal{I} \subseteq [\![d]\!]$ of size $k$ that is shattered by $\mathcal{C}$. From the definition of shattering and the construction of $\mathcal{C}$, we have that for any vector $\boldsymbol{z} \in \{0,1\}^{\mathcal{I}}$, there exists some vector $\boldsymbol{x} \in \mathcal{X}$ such that $\boldsymbol{x}[\mathcal{I}] = \boldsymbol{z}$.

We now construct the hard instance. For the simplicity of presentation, we assume that $T$ divides $|\mathcal{I}|$. We partition the $T$ rounds into $|\mathcal{I}|$ equal-length segments, each assigned to a unique element of $\mathcal{I}$. Let $i_t$ denote the unique element in $\mathcal{I}$ associated with the segment that round $t$ belongs to. The loss vector in round $t$ is then generated as

$$\boldsymbol{y}_t := \boldsymbol{e}_{i_t} \cdot \xi_t,$$

where $\xi_t$ is drawn from the Rademacher distribution, i.e., $\mathbb{P}(\xi_t = \pm 1) = 1/2$.

Now consider the expected loss incurred by the Algorithm ALG. The decisions generated by Algorithm ALG satisfy $\boldsymbol{x}_t \perp \boldsymbol{y}_t \mid \mathcal{F}_t$, i.e., $\boldsymbol{x}_t$ is conditionally independent of $\boldsymbol{y}_t$ given $\mathcal{F}_t$. Therefore,

$$\mathbb{E}\left[\sum_{t=1}^T \langle \boldsymbol{x}_t, \boldsymbol{y}_t \rangle\right] = \sum_{t=1}^T \mathbb{E}[\langle \boldsymbol{x}_t, \boldsymbol{y}_t \rangle \mid \mathcal{F}_t] = \sum_{t=1}^T \mathbb{E}[\boldsymbol{x}_t[i_t] \cdot \xi_t \mid \mathcal{F}_t] = 0.$$

On the other hand, the loss of the optimal action $\boldsymbol{x}_* \in \mathcal{X}$ satisfies

$$\min_{\boldsymbol{x}_* \in \mathcal{X}} \sum_{t=1}^T \langle \boldsymbol{x}_*, \boldsymbol{y}_t \rangle = \min_{\boldsymbol{x}_* \in \mathcal{X}} \sum_{t=1}^T \boldsymbol{x}_*[i_t] \cdot \xi_t = \min_{\boldsymbol{x}_* \in \mathcal{X}} \sum_{i \in \mathcal{I}} \boldsymbol{x}_*[i] \sum_{t:i_t=i} \xi_t.$$

According to the selection of $\mathcal{I}$, for any vector $\boldsymbol{z} \in \mathbb{R}^d$, there exists some vector $\boldsymbol{x} \in \mathcal{X}$ such that $\boldsymbol{x}[i] = 1$ if and only if $\boldsymbol{z}[i] < 0$ for every $i \in \mathcal{I}$. This implies that we can always choose some vector $\boldsymbol{x} \in \mathcal{X}$ to minimize $\langle \boldsymbol{x}[\mathcal{I}], \boldsymbol{z}[\mathcal{I}] \rangle$ simultaneously across all coordinates in $\mathcal{I}$. Thus,

$$\min_{\boldsymbol{x}_* \in \mathcal{X}} \sum_{i \in \mathcal{I}} \boldsymbol{x}_*[i] \sum_{t:i_t=i} \xi_t = \sum_{i \in \mathcal{I}} \min\left\{\sum_{t:i_t=i} \xi_t, 0\right\}.$$

Taking the expectation over the randomness of $\xi_t$, this implies

$$\mathbb{E}\left[\min_{\boldsymbol{x}_* \in \mathcal{X}} \sum_{t=1}^T \langle \boldsymbol{x}_*, \boldsymbol{y}_t \rangle\right] = \sum_{i \in \mathcal{I}} \mathbb{E}\left[\min\left\{\sum_{t:i_t=i} \xi_t, 0\right\}\right] = -\frac{1}{2} \sum_{i \in \mathcal{I}} \mathbb{E}\left|\sum_{t:i_t=i} \xi_t\right|$$
$$\leq -\sum_{i \in \mathcal{I}} \sqrt{T/(8|\mathcal{I}|)} = -\sqrt{T|\mathcal{I}|/8}.$$

where the second inequality follows from Khintchine Inequality (Lemma F.2) in which the expected absolute sum of $n$ independent Rademacher variables is at least $\sqrt{n/2}$, and the size of set $\{t : i_t = i\}$ is exactly $T/|\mathcal{I}|$.

In general, we can conclude that

$$\mathbb{E}[\text{Regret}(T)] = \mathbb{E}\left[\sum_{t=1}^{T}\langle \boldsymbol{x}_t, \boldsymbol{y}_t\rangle - \min_{\boldsymbol{x}_* \in \mathcal{X}}\sum_{t=1}^{T}\langle \boldsymbol{x}_*, \boldsymbol{y}_t\rangle\right] \geq \sqrt{\frac{T|\mathcal{I}|}{8}} = \Omega\left(\max\left\{\sqrt{\frac{T\log|\mathcal{X}|}{\log d}}, \sqrt{T}\right\}\right)$$

where the last equality follows from $|\mathcal{I}| = k = \Omega(\max\{\log|\mathcal{X}|/\log d, 1\})$.

By Yao's lemma, for any randomized algorithm ALG, there exists a sequence of loss vectors $\boldsymbol{y}_1, \boldsymbol{y}_2, \ldots, \boldsymbol{y}_T \in \mathcal{Y}$ such that the algorithm incurs a regret of at least

$$\mathbb{E}[\text{Regret}(T)] \geq \Omega\left(\max\left\{\sqrt{\frac{T\log|\mathcal{X}|}{\log d}}, \sqrt{T}\right\}\right).$$

$\square$

# B  Deferred Proofs from Section 4

**Proposition 4.1.** For any vector $\boldsymbol{z} \in \mathbb{R}^d$, the primal norm $\|\cdot\|$ is upper bounded by the $\ell_1$ and $\ell_\infty$ norms together, namely,

$$\|\boldsymbol{z}\| \leq 3\|\boldsymbol{z}\|_\infty + \frac{1}{m}\|\boldsymbol{z}\|_1.$$

*Proof.* It is sufficient to show that $\langle \boldsymbol{y}, \boldsymbol{z}\rangle \leq 3\|\boldsymbol{z}\|_\infty + (1/m)\|\boldsymbol{z}\|_1$ for any two vectors $\boldsymbol{y}, \boldsymbol{z} \in \mathbb{R}^d$ with $\|\boldsymbol{y}\|_* \leq 1$. Let $S_1 := \{i \in [d] : \boldsymbol{y}[i] \geq 0\}$ and $S_2 := \{i \in [d] : \boldsymbol{y}[i] < 0\}$. Let us assume that $|S_1| \leq d/2$.

Let us reindex the indices in $S_2$ as $\{i_1, i_2, \ldots, i_{|S_2|}\}$ such that $\boldsymbol{y}[i_1] \leq \boldsymbol{y}[i_2] \leq \ldots \leq \boldsymbol{y}[i_{|S_2|}]$. Observe that $\sum_{j=1}^{m}\boldsymbol{y}[i_j] \geq -1$. Hence, $\boldsymbol{y}[i_j] \geq -1/m$ for all $m \leq j \leq |S_2|$.

Let us reindex the indices in $S_1$ as $\{s_1, s_2, \ldots, s_{|S_1|}\}$ such that $\boldsymbol{y}[s_1] \geq \boldsymbol{y}[s_2] \geq \ldots \geq \boldsymbol{y}[s_{|S_2|}]$. Let us first consider the case when $m \leq |S_1| \leq d/2$. Observe that $\sum_{j=1}^{m}\boldsymbol{y}[s_j] \leq 1$. Hence, $\boldsymbol{y}[s_j] \leq 1/m$ for all $m \leq j \leq |S_2|$. In this case, we have the following:

$$\langle \boldsymbol{z}, \boldsymbol{y}\rangle \leq \sum_{j=1}^{m}|\boldsymbol{z}[i_j]| \cdot |\boldsymbol{y}[i_j]| + \sum_{j=m+1}^{|S_2|}|\boldsymbol{z}[i_j]| \cdot |\boldsymbol{y}[i_j]| + \sum_{j=1}^{m}|\boldsymbol{z}[s_j]| \cdot |\boldsymbol{y}[s_j]| + \sum_{j=m+1}^{|S_1|}|\boldsymbol{z}[s_j]| \cdot |\boldsymbol{y}[s_j]|$$

$$\leq \|\boldsymbol{z}\|_\infty \cdot \sum_{j=1}^{m}|\boldsymbol{y}[i_j]| + \frac{1}{m}\cdot\sum_{j=m+1}^{|S_2|}|\boldsymbol{z}[i_j]| + \|\boldsymbol{z}\|_\infty \cdot \sum_{j=1}^{m}|\boldsymbol{y}[s_j]| + \frac{1}{m}\cdot\sum_{j=m+1}^{|S_1|}|\boldsymbol{z}[s_j]|$$

$$\leq 2\|\boldsymbol{z}\|_\infty + \frac{1}{m}\|\boldsymbol{z}\|_1$$

Next, let us consider the case when $|S_1| < m$. If $|S_1| \neq 0$, then consider the set $\mathcal{I} = S_1 \cup \{i_1, i_2, \ldots, i_{m-|S_1|}\}$. Now observe that $\sum_{i \in \mathcal{I}}\boldsymbol{y}[i] \leq 1$. Hence, we have that $\sum_{j=1}^{|S_1|}\boldsymbol{y}[s_j] \leq 1 - \sum_{j=1}^{m-|S_1|}\boldsymbol{y}[i_j] \leq 2$ as $\sum_{j=1}^{m-|S_1|}\boldsymbol{y}[i_j] \geq \sum_{j=1}^{m}\boldsymbol{y}[i_j] \geq -1$. In this case, we have the following:

$$\langle \boldsymbol{z}, \boldsymbol{y}\rangle \leq \mathbb{1}\{|S_1| \neq 0\} \cdot \sum_{j=1}^{|S_1|}|\boldsymbol{z}[s_j]| \cdot |\boldsymbol{y}[i_j]| + \sum_{j=1}^{m}|\boldsymbol{z}[i_j]| \cdot |\boldsymbol{y}[i_j]| + \sum_{j=m+1}^{|S_2|}|\boldsymbol{z}[i_j]| \cdot |\boldsymbol{y}[i_j]|$$

$$\leq \mathbb{1}\{|S_1| \neq 0\} \cdot \|\boldsymbol{z}\|_\infty \cdot \sum_{j=1}^{|S_1|}|\boldsymbol{y}[s_j]| + \|\boldsymbol{z}\|_\infty \cdot \sum_{j=1}^{m}|\boldsymbol{y}[i_j]| + \frac{1}{m}\cdot\sum_{j=m+1}^{|S_2|}|\boldsymbol{z}[i_j]|$$

$$\leq 3\|\boldsymbol{z}\|_\infty + \frac{1}{m}\|\boldsymbol{z}\|_1$$

If $|S_1| > d/2$, using analogous calculations we can again show that

$$\langle \boldsymbol{z}, \boldsymbol{y} \rangle \leq 3\|\boldsymbol{z}\|_\infty + \frac{1}{m}\|\boldsymbol{z}\|_1.$$

$\square$

**Lemma 4.2.** The function $\varphi$ is $1/9$-strongly convex with respect to the primal norm $\|\cdot\|$.

*Proof.* Take $\boldsymbol{z} \in \mathbb{R}^d$. Plugging in $(a+b)^2 \leq 2a^2 + 2b^2$ for any $a, b \in \mathbb{R}$ into Proposition 4.1 implies

$$\|\boldsymbol{z}\|^2 \leq 18\|\boldsymbol{z}\|_\infty^2 + \frac{2}{m^2}\|\boldsymbol{z}\|_1^2. \tag{B.1}$$

According to the definition of the regularizer $\varphi$, the Hessian matrix of the function is a diagonal matrix with $\nabla^2\varphi(\boldsymbol{x})[i, i] = 2 + 1/(m \cdot \boldsymbol{x}[i])$. Considering the local norm of the vector $\boldsymbol{z}$ with respect to $\varphi(\boldsymbol{x})$, we have

$$\|\boldsymbol{z}\|_{\nabla^2\varphi(\boldsymbol{x})}^2 = \boldsymbol{z}^\top\nabla^2\varphi(\boldsymbol{x})\boldsymbol{z} = \sum_{i=1}^d \boldsymbol{z}[i]^2 \cdot \left(2 + \frac{1}{m\boldsymbol{x}[i]}\right) = 2\underbrace{\sum_{i=1}^d \boldsymbol{z}[i]^2}_{\mathcal{I}_1} + \frac{1}{m}\underbrace{\sum_{i=1}^d \frac{\boldsymbol{z}[i]^2}{\boldsymbol{x}[i]}}_{\mathcal{I}_2}. \tag{B.2}$$

We note that

$$\mathcal{I}_1 = \sum_{i=1}^d \boldsymbol{z}[i]^2 \geq \max_{i=1}^d \boldsymbol{z}[i]^2 = \|\boldsymbol{z}\|_\infty^2. \tag{B.3}$$

Furthermore, by the definition of the $m$-Set, we have $\sum_{i=1}^d \boldsymbol{x}[i] = m$ for $\boldsymbol{x} \in \mathcal{X}$. Therefore, we can write

$$\mathcal{I}_2 = \left(\sum_{i=1}^d \frac{\boldsymbol{z}[i]^2}{\boldsymbol{x}[i]}\right) \cdot \frac{1}{m}\left(\sum_{i=1}^d \boldsymbol{x}[i]\right) \geq \frac{1}{m}\left(\sum_{i=1}^d \sqrt{\frac{\boldsymbol{z}[i]^2}{\boldsymbol{x}[i]}} \cdot \sqrt{\boldsymbol{x}[i]}\right)^2 = \frac{1}{m}\|\boldsymbol{z}\|_1^2, \tag{B.4}$$

where the inequality follows from the Cauchy–Schwarz inequality.

Plugging (B.3) and (B.4) into (B.2), we get

$$\|\boldsymbol{z}\|_{\nabla^2\varphi(\boldsymbol{x})}^2 \geq 2\|\boldsymbol{z}\|_\infty^2 + \frac{1}{m^2}\|\boldsymbol{z}\|_1^2 \geq \frac{1}{9}\|\boldsymbol{z}\|^2,$$

where the last inequality follows from (B.1). This concludes that $\varphi$ is $1/9$-strongly convex with respect to the primal norm $\|\cdot\|$. $\square$

**Lemma 4.3.** We have $\mathcal{D}_\varphi(\boldsymbol{x}_* \| \widetilde{\boldsymbol{x}}_1) \leq m + \ln(d/m)$, for any vector $\boldsymbol{x}_* \in \mathcal{X}$.

*Proof.* From the first-order optimality of $\widetilde{\boldsymbol{x}}_1 := \mathrm{argmin}_{\boldsymbol{x}\in\mathrm{co}(\mathcal{X})} \varphi(\boldsymbol{x})$, it satisfies that for any vector $\boldsymbol{x}_* \in \mathcal{X}$, $\langle\nabla\varphi(\widetilde{\boldsymbol{x}}_1), \boldsymbol{x}_* - \widetilde{\boldsymbol{x}}_1\rangle = 0$. Therefore,

$$\mathcal{D}_\varphi(\boldsymbol{x}_* \| \widetilde{\boldsymbol{x}}_1) \leq \max_{\boldsymbol{x}\in\mathrm{co}(\mathcal{X})} \varphi(\boldsymbol{x}) - \min_{\boldsymbol{x}\in\mathrm{co}(\mathcal{X})} \varphi(\boldsymbol{x}). \tag{B.5}$$

Consider $\boldsymbol{x} \in \mathrm{co}(\mathcal{X})$. On the one hand, we have

$$\varphi(\boldsymbol{x}) = \sum_{i=1}^d \left(\boldsymbol{x}[i]^2 + \frac{1}{m}\boldsymbol{x}[i]\ln\boldsymbol{x}[i]\right) \leq \sum_{i=1}^d \boldsymbol{x}[i] + 0 = m, \tag{B.6}$$

where the first term is bounded by $\boldsymbol{x}[i]^2 \leq \boldsymbol{x}[i]$ for $\boldsymbol{x}[i] \in [0, 1]$, and the second term satisfies $\boldsymbol{x}[i]\ln\boldsymbol{x}[i] \leq 0$.

On the other hand, define $\boldsymbol{p} = \boldsymbol{x}/m$. Then, $\boldsymbol{p}$ lies in the probability simplex by the definition of $\boldsymbol{x} \in \mathrm{co}(\mathcal{X})$.

$$\varphi(\boldsymbol{x}) = \sum_{i=1}^{d} \left( \boldsymbol{x}[i]^2 + \frac{1}{m} \boldsymbol{x}[i] \ln \boldsymbol{x}[i] \right)$$

$$\geq 0 + \sum_{i=1}^{d} \boldsymbol{p}[i] \ln \boldsymbol{p}[i] + \sum_{i=1}^{d} \boldsymbol{p}[i] \ln m$$

$$= \sum_{i=1}^{d} \boldsymbol{p}[i] \ln \boldsymbol{p}[i] + \ln m$$

$$\geq -\ln(d/m), \tag{B.7}$$

where the first inequality uses $\boldsymbol{x}[i]^2 \geq 0$, and the second inequality follows from the entropy upper bound over the probability simplex.

Finally, plugging (B.6) and (B.7) into (B.5) yields

$$\mathcal{D}_\varphi(\boldsymbol{x}_* \,\|\, \widetilde{\boldsymbol{x}}_1) \leq m + \ln(d/m).$$

$\square$

**Theorem 4.5.** Consider integers $m, d, T$ such that $1 \leq m \leq d/2 \leq \exp(T/3)/2$. For the $m$-sets problem and any Algorithm ALG, there exists a sequence of loss vectors $\boldsymbol{y}_1, \boldsymbol{y}_2, \ldots, \boldsymbol{y}_T \in \mathcal{Y}$ such that the algorithm incurs a expected regret of at least

$$\mathbb{E}[\mathrm{Regret}(T)] = \mathbb{E}\left[ \sum_{t=1}^{T} \langle \boldsymbol{x}_t, \boldsymbol{y}_t \rangle - \min_{\boldsymbol{x}_* \in \mathcal{X}} \sum_{t=1}^{T} \langle \boldsymbol{x}_*, \boldsymbol{y}_t \rangle \right] \geq \Omega\left( \sqrt{Tm + T\log(d/m)} \right).$$

*Proof.* For every vector $\boldsymbol{x} \in \mathcal{X}$, we construct a corresponding indicator set $C_{\boldsymbol{x}} := \{i \in \llbracket d \rrbracket : \boldsymbol{x}[i] = 1\}$. Denote by $\mathcal{C} := \{C_{\boldsymbol{x}} : \boldsymbol{x} \in \mathcal{X}\}$ the collection of these sets. We have $|\mathcal{C}| = |\mathcal{X}|$. Notice that the set $\mathcal{I} = \{1, \ldots, m\}$, of size $m$, is shattered by $\mathcal{C}$. Hence, by the same argument as in the proof of Theorem 3.2, for any randomized algorithm ALG there exists a sequence of loss vectors $\boldsymbol{y}_1, \boldsymbol{y}_2, \ldots, \boldsymbol{y}_T \in \mathcal{Y}$ such that the algorithm incurs a regret of at least $\Omega(\sqrt{Tm})$.

To prove that $\mathrm{Regret}(T) \geq \Omega\left(\sqrt{T\log(d/m)}\right)$, we construct a hard instance as follows. For simplicity of presentation, we assume that $d$ divides $m$. Let $K := d/m$. Let $\mathcal{D}_K$ denote the zero-mean distribution over $\{-1, +1\}^K$ from Lemma F.4. In each round $t$, the environment first draws a vector $\boldsymbol{z}_t \sim \mathcal{D}_K$ and assigns

$$\boldsymbol{y}_t := \left[ \underbrace{\frac{\boldsymbol{z}_t[1]}{m}, \frac{\boldsymbol{z}_t[1]}{m}, \ldots, \frac{\boldsymbol{z}_t[1]}{m}}_{m}, \underbrace{\frac{\boldsymbol{z}_t[2]}{m}, \frac{\boldsymbol{z}_t[2]}{m}, \ldots, \frac{\boldsymbol{z}_t[2]}{m}}_{m}, \ldots, \underbrace{\frac{\boldsymbol{z}_t[K]}{m}, \frac{\boldsymbol{z}_t[K]}{m}, \ldots, \frac{\boldsymbol{z}_t[K]}{m}}_{m} \right]^{\top}.$$

Next for all $i \in \llbracket K \rrbracket$, let $\boldsymbol{x}^{(i)}$ be a vector in $\mathcal{X}$ that is defined as

$$\boldsymbol{x}^{(i)} := \left[ 0, 0, \ldots, 0, \underbrace{1, 1, \ldots, 1}_{i\text{-th block of } m \text{ coordinates}}, 0, 0, \ldots, 0 \right]^{\top},$$

that is, $\boldsymbol{x}^{(i)}[j] = 1$ if $(i-1)m + 1 \leq j \leq i \cdot m$ and $\boldsymbol{x}^{(i)}[j] = 0$ otherwise.

We begin by considering a deterministic algorithm ALG and establish a regret lower bound for it. The result is then extended to randomized algorithms via Yao's lemma. Let $\boldsymbol{x}_t$ be the vector chosen by the Algorithm ALG in the round $t$. Observe that the expected loss of ALG is zero as $\mathbb{E}[\langle \boldsymbol{x}_t, \boldsymbol{y}_t \rangle] = \mathbb{E}[\mathbb{E}[\langle \boldsymbol{x}_t, \boldsymbol{y}_t \rangle | \mathcal{F}_t]] = \mathbb{E}[\langle \boldsymbol{x}_t, \mathbb{E}[\boldsymbol{y}_t | \mathcal{F}_t] \rangle] = 0$. On the other hand, we have

$$\mathbb{E}\left[ \min_{\boldsymbol{x} \in \mathcal{X}} \sum_{t=1}^{T} \langle \boldsymbol{x}, \boldsymbol{y}_t \rangle \right] \leq \mathbb{E}\left[ \min_{i \in \llbracket K \rrbracket} \sum_{t=1}^{T} \langle \boldsymbol{x}^{(i)}, \boldsymbol{y}_t \rangle \right] = \mathbb{E}\left[ \min_{i \in \llbracket K \rrbracket} \sum_{t=1}^{T} \langle \boldsymbol{e}_i, \boldsymbol{z}_t \rangle \right].$$

Hence, the regret of ALG is lower bounded as

$$\mathbb{E}[\text{Regret}(T)] = \mathbb{E}\left[\sum_{t=1}^{T}\langle \boldsymbol{x}_t, \boldsymbol{y}_t\rangle - \min_{\boldsymbol{x}\in\mathcal{X}}\sum_{t=1}^{T}\langle \boldsymbol{x}, \boldsymbol{y}_t\rangle\right] \geq -\mathbb{E}\left[\min_{i\in\llbracket K\rrbracket}\sum_{t=1}^{T}\langle \boldsymbol{e}_i, \boldsymbol{z}_t\rangle\right] \geq \Omega(\sqrt{T\log(d/m)}),$$

where the last inequality follows from Lemma F.4.

By Yao's lemma, for any randomized algorithm ALG, there exists a sequence of loss vectors $\boldsymbol{y}_1, \boldsymbol{y}_2, \ldots, \boldsymbol{y}_T \in \mathcal{Y}$ such that the algorithm incurs a regret of at least $\Omega\left(\sqrt{T\log(d/m)}\right)$.

Combining both the lower bounds, we conclude the desired regret lower bound:

$$\mathbb{E}[\text{Regret}(T)] \geq \Omega\left(\max\left\{\sqrt{Tm}, \sqrt{T\log(d/m)}\right\}\right) \geq \Omega\left(\sqrt{Tm + T\log(d/m)}\right).$$

$\square$

**Theorem 4.6.** Consider integers $m, d, T$ such that $1 \leq m \leq d/2 \leq \exp(T/3)/2$ and $m\log(d/m) \leq T$. For any $\eta > 0$, there exists a there exists a sequence of loss vectors $\boldsymbol{y}_1, \boldsymbol{y}_2, \ldots, \boldsymbol{y}_T \in \mathcal{Y}$ over $m$-sets such that the HEDGE algorithm with learning rate $\eta$ incurs a expected regret of at least

$$\mathbb{E}[\text{Regret}(T)] = \mathbb{E}\left[\sum_{t=1}^{T}\langle \boldsymbol{x}_t, \boldsymbol{y}_t\rangle - \min_{\boldsymbol{x}_*\in\mathcal{X}}\sum_{t=1}^{T}\langle \boldsymbol{x}_*, \boldsymbol{y}_t\rangle\right] \geq \Omega\left(\sqrt{Tm\log(d/m)}\right).$$

*Proof.* Recall that the HEDGE algorithm with learning rate $\eta > 0$ selects an action $\boldsymbol{x} \in \mathcal{X}$ in round $t$ with probability proportional to:

$$\mathbb{P}_t(\boldsymbol{x}) \propto w_t(\boldsymbol{x}) := \exp\left(-\eta\sum_{\tau=1}^{t-1}\langle \boldsymbol{x}, \boldsymbol{y}_\tau\rangle\right).$$

If $m \leq 20$, the regret lower bound of $\Omega\left(\sqrt{mT\ln(d/m)}\right) = \Omega\left(\sqrt{T\log d}\right)$ for HEDGE directly follows from Theorem 4.5. Hence, for the rest of the proof we assume that $m \geq 20$.

We divide the proof into two cases based on how $\eta$ compares to the base learning rate

$$\eta_0 := \sqrt{\frac{m\ln(d/m)}{T}} = \Theta\left(\sqrt{\frac{\log|\mathcal{X}|}{T}}\right).$$

**Regime where the learning rate is small, $\eta \leq \eta_0$:**

When the learning rate is small, we construct the hard instance by assigning a fixed loss vector $\boldsymbol{y}_t$ across the rounds according to

$$\boldsymbol{y}_t[i] := (1/m) \cdot \mathbb{1}[i \in \llbracket m\rrbracket].$$

Let the set of bad actions that place small Hamming weight on the first $m$ coordinates be defined as

$$\mathcal{S} := \left\{\boldsymbol{x} \in \mathcal{X} : \left|\{i \in \llbracket m\rrbracket : \boldsymbol{x}_t[i] \neq 1\}\right| \geq \left\lfloor\frac{m}{20}\right\rfloor\right\}.$$

From the calculation in Lemma G.3, we have that the subset $\mathcal{S}$ is relatively large with respect to the whole decision set $\mathcal{X}$:

$$\frac{|\mathcal{X}\setminus\mathcal{S}|}{|\mathcal{X}|} \leq \exp\left(-\frac{m}{20}\cdot\ln\left(\frac{d}{m}\right)\right) =: w_0.$$

For any round $t \leq t_0 := \sqrt{(m/400)\cdot T\ln(d/m)}$, the weight of any action $\boldsymbol{x} \in \mathcal{S}$ is at least

$$w_t(\boldsymbol{x}) = \exp\left(-\eta\sum_{\tau=1}^{t-1}\langle \boldsymbol{x}, \boldsymbol{y}_\tau\rangle\right) \geq \exp(-\eta_0\cdot t_0) = \exp\left(-\frac{m}{20}\cdot\ln\left(\frac{d}{m}\right)\right) = w_0.$$

Furthermore, for any $\boldsymbol{x} \in \mathcal{X}\setminus\mathcal{S}$, it is clear that the weight satisfies $w_t(\boldsymbol{x}) \leq 1$. Now consider the probability that some bad action $\boldsymbol{x}_t \in \mathcal{S}$ is chosen in round $t \leq t_0$. We have:

$$\mathbb{P}(\boldsymbol{x}_t \in \mathcal{S}) = \frac{\sum_{\boldsymbol{x}\in\mathcal{S}}w_t(\boldsymbol{x})}{\sum_{\boldsymbol{x}\in\mathcal{X}}w_t(\boldsymbol{x})} \geq \left(1+\frac{\sum_{\boldsymbol{x}\in\mathcal{X}\setminus\mathcal{S}}w_t(\boldsymbol{x})}{\sum_{\boldsymbol{x}\in\mathcal{S}}w_t(\boldsymbol{x})}\right)^{-1} \geq \left(1+\frac{|\mathcal{X}\setminus\mathcal{S}|}{|\mathcal{S}|}\cdot\frac{\max_{\boldsymbol{x}\in\mathcal{X}}w_t(\boldsymbol{x})}{\min_{\boldsymbol{x}\in\mathcal{S}}w_t(\boldsymbol{x})}\right)^{-1} \geq \frac{1}{2}.$$

Observe that in round $t$, if HEDGE chooses any action $\boldsymbol{x} \in \mathcal{S}$, then it incurs a regret of at least $(1/m) \cdot \lfloor m/20 \rfloor \geq 1/40$, where the last inequality follows from Lemma G.1. Hence, HEDGE incurs a constant regret in each round $t \leq t_0$. Note that as loss vector is fixed, HEDGE can not achieve negative regret in any round. This indicates that HEDGE incurs a total regret of at least

$$\mathbb{E}[\text{Regret}(T)] \geq \min\{t_0, T\} \cdot \mathbb{P}(\boldsymbol{x}_t \in \mathcal{S}) \cdot \frac{1}{40} \geq \Omega\left(\sqrt{Tm\log(d/m)}\right),$$

for $m\log(d/m) \leq T$, when the learning rate $\eta \leq \eta_0$ is small.

**Regime where the learning rate is large, $\eta > \eta_0$:**

When the learning rate is large, we construct the hard instance as follows. We set $\boldsymbol{y}_t[i] = 0$ for all coordinates $i \geq 2$ across all rounds $t \in [\![T]\!]$. For the first coordinate, we define $\boldsymbol{y}_t[1]$ in two phases, determined by $t_0 := \ln(d/m)/\eta \leq \sqrt{T\ln(d/m)/m}$.

- **Phase 1:** For the first $t \leq \lfloor t_0 \rfloor$ rounds, we assign $\boldsymbol{y}_t[1] = -1$. If $t_0 \notin \mathbb{N}$, then in round $t = \lceil t_0 \rceil$, we assign $\boldsymbol{y}_t[1] = -(t_0 - \lfloor t_0 \rfloor)$. In this way, the cumulative loss over the first $\lceil t_0 \rceil$ rounds is exactly $-t_0$ on the first coordinate.

- **Phase 2:** For all remaining rounds $t > \lceil t_0 \rceil$, we assign $\boldsymbol{y}_t[1]$ in an alternating manner as follows:

$$\boldsymbol{y}_t[1] := \begin{cases} +1 & \text{if } t - \lceil t_0 \rceil \text{ is odd,} \\ -1 & \text{if } t - \lceil t_0 \rceil \text{ is even.} \end{cases}$$

Let $\mathcal{S}_0 := \{\boldsymbol{x} \in \mathcal{X} : \boldsymbol{x}[1] = 0\}$ and $\mathcal{S}_1 := \{\boldsymbol{x} \in \mathcal{X} : \boldsymbol{x}[1] = 1\}$ denote a partition of the action set based on whether mass is placed on the first coordinate. Note that

$$|\mathcal{S}_0| = \left(1 - \frac{m}{d}\right) \cdot \binom{d}{m}, \qquad |\mathcal{S}_1| = \frac{m}{d} \cdot \binom{d}{m}.$$

According to the loss structure of the hard instance, every action in $\mathcal{S}_1$ incurs a same loss of $\boldsymbol{y}_t[1]$ in each round $t$, while every action in $\mathcal{S}_0$ incurs zero loss. According to the HEDGE update rule, one can see running HEDGE on $\mathcal{X}$ is essentially equivalent to running the algorithm on two aggregate actions $\boldsymbol{x_0}$ and $\boldsymbol{x_1}$, where $\boldsymbol{x_0}$ represents playing a uniformly random action from $\mathcal{S}_0$ and $\boldsymbol{x_1}$ represents playing uniformly from $\mathcal{S}_1$. The initial weights are given by $w_1(\boldsymbol{x_0}) = |\mathcal{S}_0|$ and $w_1(\boldsymbol{x_1}) = |\mathcal{S}_1|$. In each round $t$, action $\boldsymbol{x_1}$ suffers a loss of $\boldsymbol{y}_t[1]$, and action $\boldsymbol{x_0}$ suffers a loss of $0$.

We analyze HEDGE on actions $\boldsymbol{x_0}$ and $\boldsymbol{x_1}$. For simplicity, we assume that $T - \lceil t_0 \rceil$ is an even number. Consider a round $t_1 := \lceil t_0 \rceil + 2b$, for some integer $b \geq 0$. Let $w^{\boldsymbol{0}}$ and $w^{\boldsymbol{1}}$ denote the weights of actions $\boldsymbol{x_0}$ and $\boldsymbol{x_1}$ in round $t_1 + 1$, respectively. Then, HEDGE chooses action $\boldsymbol{x_1}$ with probability $\frac{w^{\boldsymbol{1}}}{w^{\boldsymbol{0}} + w^{\boldsymbol{1}}}$ in round $t_1 + 1$, and with probability $\frac{w^{\boldsymbol{1}} \cdot \exp(-\eta)}{w^{\boldsymbol{0}} + w^{\boldsymbol{1}} \cdot \exp(-\eta)}$ in round $t_1 + 2$. Therefore, the total expected loss incurred in these two rounds is

$$\sum_{t=t_1+1}^{t_1+2} \mathbb{E}[\langle \boldsymbol{x}_t, \boldsymbol{y}_t \rangle] = (+1) \cdot \frac{w^{\boldsymbol{1}}}{w^{\boldsymbol{0}} + w^{\boldsymbol{1}}} + (-1) \cdot \frac{w^{\boldsymbol{1}} \cdot \exp(-\eta)}{w^{\boldsymbol{0}} + w^{\boldsymbol{1}} \cdot \exp(-\eta)} = \frac{w^{\boldsymbol{0}} w^{\boldsymbol{1}} (1 - e^{-\eta})}{(w^{\boldsymbol{0}} + w^{\boldsymbol{1}})(w^{\boldsymbol{0}} + w^{\boldsymbol{1}} \cdot e^{-\eta})}.$$

Since the cumulative loss in the first $t_1$ rounds of action $\boldsymbol{x_0}$ is $0$, and that of action $\boldsymbol{x_1}$ is $-t_0$, we have

$$w^{\boldsymbol{0}} = |\mathcal{S}_0| \cdot \exp(-\eta \cdot 0) = |\mathcal{S}_0| = \left(1 - \frac{m}{d}\right) \cdot \binom{d}{m},$$

$$w^{\boldsymbol{1}} = |\mathcal{S}_1| \cdot \exp(-\eta \cdot (-t_0)) = |\mathcal{S}_1| \cdot \exp(\ln(d/m)) = \binom{d}{m}.$$

From $1 \leq m \leq d/2$, it follows that $1 \leq w^{\boldsymbol{1}}/w^{\boldsymbol{0}} \leq 2$. Plugging this into the expression gives:

$$\sum_{t=t_1+1}^{t_1+2} \mathbb{E}[\langle \boldsymbol{x}_t, \boldsymbol{y}_t \rangle] \geq \frac{w^{\boldsymbol{0}} w^{\boldsymbol{1}} (1 - e^{-\eta})}{(w^{\boldsymbol{0}} + w^{\boldsymbol{1}})^2} \geq \Omega\left(\min\{\eta, 1\}\right),$$

where the last inequality follows from $1 \leq w^{\boldsymbol{1}}/w^{\boldsymbol{0}} \leq 2$ and also $(1 - e^{-x}) \geq (1 - e^{-1}) \cdot \min\{x, 1\}$ for all $x > 0$.

Now observe that any action $\boldsymbol{x} \in \mathcal{S}_1$ is the best action in hindsight over $T$ rounds, incurring a cumulative loss of $-t_0$. The HEDGE algorithm suffers at least $-t_0$ expected loss in the first phase and incurs an expected loss of at least $\Omega(\min\{\eta, 1\})$ every two rounds in the second phase. Therefore, the total regret incurred by HEDGE is at least

$$\mathbb{E}[\mathrm{Regret}(T)] \geq \frac{T - \lceil t_0 \rceil}{2} \cdot \Omega\big(\min\{\eta, 1\}\big) \geq \Omega\big(\sqrt{Tm \log(d/m)}\big),$$

for $m \log(d/m) \leq T$, when the learning rate $\eta > \eta_0$ is large. $\qquad\square$

## C    Deferred Proof from Section 5

**Theorem 5.1.** Consider any instance of the online multitask learning problem on $\mathcal{X}$ of dimension $d \geq 2$. Let the corresponding loss vector set $\mathcal{Y}$ satisfy Assumption 2.1. Consider an integer $T \geq 3 \log |\mathcal{X}|$. Then for any Algorithm ALG, there exists a sequence of loss vectors $\boldsymbol{y}_1, \boldsymbol{y}_2, \ldots, \boldsymbol{y}_T \in \mathcal{Y}$ such that the algorithm incurs a regret of at least

$$\mathbb{E}[\mathrm{Regret}(T)] = \mathbb{E}\left[\sum_{t=1}^{T} \langle \boldsymbol{x}_t, \boldsymbol{y}_t \rangle - \min_{\boldsymbol{x}_* \in \mathcal{X}} \sum_{t=1}^{T} \langle \boldsymbol{x}_*, \boldsymbol{y}_t \rangle\right] \geq \Omega\big(\sqrt{T \log |\mathcal{X}|}\big).$$

*Proof.* We begin by considering a deterministic algorithm ALG and establish a regret lower bound for it. The result is then extended to randomized algorithms via Yao's lemma.

Let us assume that there are $m$ expert problems and the $i$-th problem has $d_i \geq 2$ experts. Recall that $d_{1:i} = \sum_{j=1}^{i} d_j$ and $d_{1:0} = 0$. For all $i \in [\![m]\!]$, let $T_i := \frac{\log d_i}{\sum_{j=1}^{m} \log d_j} \cdot T$. For simplicity of presentation, let us assume that $T_i$ is a positive integer for all $i \in [\![m]\!]$. Let $T_{1:i} = \sum_{j=1}^{i} T_j$ and $d_{1:0} = 0$. For any $K \geq 2$, let $\mathcal{D}_K$ be the zero-mean distribution over $\{-1, +1\}^K$ from Lemma F.4.

We begin by dividing the $T$ rounds into $m$ phases, where $i$-th phase lasts for $T_i$ rounds. In a round $t$ in the $i$-th phase, we choose $\boldsymbol{z}_t \sim \mathcal{D}_{d_i}$ and define the loss vector $\boldsymbol{y}_t$ as follows:

$$\boldsymbol{y}_t[s] = \begin{cases} \boldsymbol{z}_t[j] & \text{if } s = d_{1:i-1} + j, \\ 0 & \text{otherwise.} \end{cases}$$

Let $\boldsymbol{x}_t$ be the vector chosen by the Algorithm ALG in the round $t$. Observe that the expected loss of ALG is zero as $\mathbb{E}[\langle \boldsymbol{x}_t, \boldsymbol{y}_t \rangle] = \mathbb{E}[\mathbb{E}[\langle \boldsymbol{x}_t, \boldsymbol{y}_t \rangle | \mathcal{F}_t]] = 0$. Given the structure of the loss vector $\boldsymbol{y}_t$, the problem reduces to solving $m$ separate expert problems, each in its own phase. Consequently, the regret incurred by ALG can be decomposed as follows:

$$\begin{aligned}
\mathbb{E}\left[\mathrm{Regret}(T)\right] &= \sum_{i=1}^{m} -\mathbb{E}\left[\min_{j \in [\![d_i]\!]} \sum_{t=T_{1:i-1}+1}^{T_{1:i}} \langle \boldsymbol{e}_j, \boldsymbol{z}_t \rangle\right] \\
&\geq c_0 \cdot \sum_{i=1}^{m} \sqrt{T_i \log d_i} && \text{(due to Lemma F.4)} \\
&= c_0 \cdot \frac{(\sum_{i=1}^{m} \log d_i) \cdot \sqrt{T}}{\sqrt{\sum_{i=1}^{m} \log d_i}} \\
&= c_0 \cdot \sqrt{T \log |\mathcal{X}|}, && \text{(as } \prod_{i=1}^{m} d_i = |\mathcal{X}|)
\end{aligned}$$

where $c_0$ is the universal constant from Lemma F.4.

By Yao's lemma, for any randomized algorithm ALG, there exists a sequence of loss vectors $\boldsymbol{y}_1, \boldsymbol{y}_2, \ldots, \boldsymbol{y}_T \in \mathcal{Y}$ such that the algorithm incurs a regret of at least $\Omega\big(\sqrt{T \log |\mathcal{X}|}\big)$. $\qquad\square$

## D    Deferred Proof from Section 6

**Theorem 6.1.** For any integers $d, N, T$ such that $16 \leq 2d \leq N \leq 2^d$ and $3 \log N \leq T$, and for any algorithm ALG, there exists a DAG $G$ with at most $d$ edges and at most $N$ paths from source s to sink

t, and a corresponding sequence of loss vectors $\boldsymbol{y}_1, \boldsymbol{y}_2, \ldots, \boldsymbol{y}_T \in \mathcal{Y}$ such that the algorithm incurs a regret lower bound of

$$\mathbb{E}[\text{Regret}(T)] = \mathbb{E}\left[\sum_{t=1}^{T} \langle \boldsymbol{x}_t, \boldsymbol{y}_t \rangle - \min_{\boldsymbol{x}_* \in \mathcal{X}} \sum_{t=1}^{T} \langle \boldsymbol{x}_*, \boldsymbol{y}_t \rangle\right] \geq \Omega(\sqrt{T \log N}).$$

*Proof.* Let us assume that $d \geq 8$ and $2d \leq N \leq 2^d$. Let $d_0 := 8\lfloor d/8 \rfloor$. Due to Lemma G.1, we have $d/2 \leq d_0 \leq d$. Let $N_0 := \min\{N, 2^{d_0/4}\}$. Note that $\log(N_0) = \Theta(\log N)$. Let $m$ be the largest integer such that $m \leq d_0/8$ and $(\frac{d_0}{2m})^m \leq N_0$. Now we claim that $m\log(\lfloor \frac{d_0}{2m} \rfloor) \geq \Omega(\log N)$.

If $m = d_0/8$, then we have $m\log(\lfloor \frac{d_0}{2m} \rfloor) = (d_0/8)\log(4) \geq \Omega(\log N)$. Let us consider the case when $m < d_0/8$. Due to Lemma G.2, $(a/x)^x$ is increasing in the range $[1, a/3]$ for any $a > 3$. Hence, we have $(\frac{d_0}{2m})^m \leq N_0 < (\frac{d_0}{2(m+1)})^{m+1}$. Next we have the following:

$$\frac{(m+1)\log(\frac{d_0}{2(m+1)})}{m\log(\frac{d_0}{2m})} \leq \frac{(m+1)\log(\frac{d_0}{2m})}{m\log(\frac{d_0}{2m})} \leq \frac{m+1}{m} \leq 2$$

Due to the above inequality, Lemma G.1 and $d_0/m \geq 8$, we have the following

$$m\log\left(\left\lfloor \frac{d_0}{2m} \right\rfloor\right) \geq m\log\left(\frac{d_0}{4m}\right) \geq \frac{m}{2}\log\left(\frac{d_0}{2m}\right) \geq \frac{m+1}{4}\log\left(\frac{d_0}{2(m+1)}\right) \geq \Omega(\log N).$$

Consider a DAG $G = (V, E)$ where the set of vertices $V$ and the set of edges $E$ are defined as follows:

$$V := \{v_i\}_{i=0}^{m} \cup \left\{ v_i^j \mid i \in [\![m]\!], j \in \left[\!\!\left[\left\lfloor \frac{d_0}{2m} \right\rfloor\right]\!\!\right] \right\}, \quad E := \left\{ (v_{i-1}, v_i^j), (v_i^j, v_i) \mid i \in [\![m]\!], j \in \left[\!\!\left[\left\lfloor \frac{d_0}{2m} \right\rfloor\right]\!\!\right] \right\}$$

Observe that $|E| = 2m \cdot \left[\!\!\left[\left\lfloor \frac{d_0}{2m} \right\rfloor\right]\!\!\right] \leq d$. Also observe that the number of paths from source to sink in $G$ is $\left\lfloor \frac{d_0}{2m} \right\rfloor^m \leq N_0 \leq N$. Hence, $G$ is in the family of DAGs $\mathcal{F}_{d,N}$. Now we show that any algorithm incurs a regret of $\Omega(\sqrt{T \log N})$ on the DAG $G$.

For any $K \geq 2$, let $\mathcal{D}_K$ denote the zero-mean distribution over $\{-1, +1\}^K$ from Lemma F.4. We begin by dividing the $T$ rounds into $m + 1$ phases: the first $m$ phases each last for $\lfloor T/m \rfloor$ rounds, and the $(m + 1)$-th phase lasts for the remaining $T - m\lfloor T/m \rfloor$ rounds. In any round $t$ of the $i$-th phase with $i \leq m$, we sample $\boldsymbol{z}_t \sim \mathcal{D}_{\lfloor d_0/(2m) \rfloor}$, and define the loss vector $\boldsymbol{y}_t$ as follows:

$$\boldsymbol{y}_t[e] = \begin{cases} \boldsymbol{z}_t[j] & \text{if } e = (v_{i-1}, v_i^j), \\ 0 & \text{otherwise}. \end{cases}$$

In a round $t$ in the $m + 1$-th phase, we choose $\boldsymbol{y}_t = \boldsymbol{0}$.

We begin by considering a deterministic algorithm ALG and establish a regret lower bound for it. The result is then extended to randomized algorithms via Yao's lemma. Let $\boldsymbol{x}_t$ be the vector chosen by the Algorithm ALG in the round $t$. Observe that the expected loss of ALG is zero as $\mathbb{E}[\langle \boldsymbol{x}_t, \boldsymbol{y}_t \rangle] = \mathbb{E}[\mathbb{E}[\langle \boldsymbol{x}_t, \boldsymbol{y}_t \rangle | \mathcal{F}_t]] = 0$. Given the structure of the loss vector $\boldsymbol{y}_t$, the problem reduces to solving $m$ separate expert problems, one in each phase $i \leq m$. Consequently, the regret incurred by ALG can be decomposed as follows:

$$\begin{aligned}
\mathbb{E}[\text{Regret}(T)] &= \sum_{i=1}^{m} -\mathbb{E}\left[ \min_{j \in [\![\lfloor d_0/(2m) \rfloor]\!]} \sum_{t=(i-1)\cdot\lfloor T/m \rfloor + 1}^{i \cdot \lfloor T/m \rfloor} \langle \boldsymbol{e}_j, \boldsymbol{z}_t \rangle \right] \\
&\geq (c_0/2) \cdot \sum_{i=1}^{m} \sqrt{(T/m)\log(\lfloor d_0/(2m) \rfloor)} \quad \text{(due to Lemma F.4 and Lemma G.1)} \\
&= (c_0/2) \cdot \sqrt{mT \log(\lfloor d_0/(2m) \rfloor)} \\
&= \Omega\left(\sqrt{T \log N}\right), \quad \text{(as } m\log(\lfloor d_0/(2m) \rfloor) \geq \Omega(\log N))
\end{aligned}$$

where $c_0$ is the universal constant from Lemma F.4.

By Yao's lemma, for any randomized algorithm ALG, there exists a sequence of loss vectors $\boldsymbol{y}_1, \boldsymbol{y}_2, \ldots, \boldsymbol{y}_T \in \mathcal{Y}$ such that the algorithm incurs a regret of at least $\Omega\left(\sqrt{T \log N}\right)$. $\qquad\square$

**Lemma 6.2.** For any $\widetilde{\boldsymbol{x}} \in \mathrm{co}(\mathcal{X})$, we have

$$\psi(\widetilde{\boldsymbol{x}}) = -H(\boldsymbol{x}) := \underset{\boldsymbol{x} \sim \mathcal{D}(\widetilde{\boldsymbol{x}})}{\mathbb{E}} [\ln \mathbb{P}(\boldsymbol{x})],$$

where $H(\cdot)$ denotes the Shannon entropy of the random variable.

*Proof.* From the procedure of Markovian sampling in Section 6.1, we have for any $\boldsymbol{x} \in \mathcal{X}$

$$\mathbb{P}(\boldsymbol{x}) = \prod_{(u,v) \in E : \boldsymbol{x}[(u,v)]=1} \frac{\widetilde{\boldsymbol{x}}[(u,v)]}{\widetilde{\boldsymbol{x}}[u]} = \frac{\prod_{e \in E : \boldsymbol{x}[e]=1} \widetilde{\boldsymbol{x}}[e]}{\prod_{v \in V : \boldsymbol{x}[v]=1} \widetilde{\boldsymbol{x}}[v]}.$$

Therefore, we have

$$-H(\boldsymbol{x}) = \underset{\boldsymbol{x} \sim \mathcal{D}(\widetilde{\boldsymbol{x}})}{\mathbb{E}} [\ln \mathbb{P}(\boldsymbol{x})]$$

$$= \underset{\boldsymbol{x} \sim \mathcal{D}(\widetilde{\boldsymbol{x}})}{\mathbb{E}} \left[ \sum_{e \in E : \boldsymbol{x}[e]=1} \ln \widetilde{\boldsymbol{x}}[e] - \sum_{v \in V : \boldsymbol{x}[v]=1} \ln \widetilde{\boldsymbol{x}}[v] \right]$$

$$= \underset{\boldsymbol{x} \sim \mathcal{D}(\widetilde{\boldsymbol{x}})}{\mathbb{E}} \left[ \sum_{e \in E} \mathbb{1}\{\boldsymbol{x}[e] = 1\} \cdot \ln \widetilde{\boldsymbol{x}}[e] - \sum_{v \in V} \mathbb{1}\{\boldsymbol{x}[v] = 1\} \cdot \ln \widetilde{\boldsymbol{x}}[v] \right]$$

$$= \sum_{e \in E} \widetilde{\boldsymbol{x}}[e] \ln \widetilde{\boldsymbol{x}}[e] - \sum_{v \in V} \widetilde{\boldsymbol{x}}[v] \ln \widetilde{\boldsymbol{x}}[v] = \psi(\widetilde{\boldsymbol{x}}),$$

where the fourth equality follows from $\mathbb{E}[\boldsymbol{x}] = \widetilde{\boldsymbol{x}}$, which concludes the proof for the equality. $\qquad\square$

**Lemma 6.3.** The function $\psi$ is $1/10$-strongly convex with respect to the primal norm $\|\cdot\|$ in $\mathrm{span}(\mathcal{X})$.

*Proof.* It is sufficient to show that $10\|\boldsymbol{z}\|_{\nabla^2 \psi(\boldsymbol{x})}^2 \geq \langle \boldsymbol{y}, \boldsymbol{z} \rangle^2$ for any vector $\boldsymbol{x} \in \mathrm{relint}(\mathrm{co}(\mathcal{X}))$, $\|\boldsymbol{y}\|_* \leq 1$, and $\boldsymbol{z} \in \mathrm{span}(\mathcal{X})$.

Consider a weight assignment $\boldsymbol{y}$ of the edges. For every vertex $v \in V$, we define $d_{\min}(v) \in \mathbb{R}$ as the weight of the shortest path from $v$ to $\mathsf{t}$, and $d_{\max}(v) \in \mathbb{R}$ as the weight of the longest path from $v$ to $\mathsf{t}$. We have $d_{\min}(\mathsf{t}) = d_{\max}(\mathsf{t}) = 0$, as well as $-1 \leq d_{\min}(\mathsf{s}) \leq d_{\max}(\mathsf{s}) \leq 1$.

We construct a vector $\boldsymbol{y}'$ by assigning

$$\boldsymbol{y}'[(u,v)] = \boldsymbol{y}[(u,v)] + d_{\min}(v) - d_{\min}(u)$$

for every edge $(u, v) \in E$. By the triangle inequality, all values $\boldsymbol{y}'[(u,v)]$ are nonnegative. Since $\boldsymbol{z} \in \mathrm{span}(\mathcal{X})$, we have

$$\langle \boldsymbol{y}', \boldsymbol{z} \rangle = \sum_{(u,v) \in E} \boldsymbol{z}[(u,v)] \cdot \big( \boldsymbol{y}[(u,v)] + d_{\min}(v) - d_{\min}(u) \big)$$

$$= \sum_{e \in E} \boldsymbol{z}[e] \cdot \boldsymbol{y}[e] + \sum_{v \in V \setminus \{\mathsf{s},\mathsf{t}\}} d_{\min}(v) \cdot \left( \sum_{e \in \delta^-(v)} \boldsymbol{z}[e] - \sum_{e \in \delta^+(v)} \boldsymbol{z}[e] \right) - d_{\min}(\mathsf{s}) \cdot \sum_{e \in \delta^+(\mathsf{s})} \boldsymbol{z}[e]$$

$$= \langle \boldsymbol{y}, \boldsymbol{z} \rangle - d_{\min}(\mathsf{s}) \cdot \boldsymbol{z}[\mathsf{s}], \tag{D.1}$$

where $\boldsymbol{z}[\mathsf{s}] := \sum_{e \in \delta^+(\mathsf{s})} \boldsymbol{z}[e]$ and the last equality follows from the flow constraints implied by $\boldsymbol{z} \in \mathrm{span}(\mathcal{X})$. Note that this equality also holds for $\boldsymbol{x} \in \mathrm{relint}(\mathrm{co}(\mathcal{X}))$. Combined with the facts that $\langle \boldsymbol{y}, \boldsymbol{x} \rangle \leq 1$, $d_{\min}(\mathsf{s}) \geq -1$, and $\boldsymbol{x}[\mathsf{s}] = 1$, we have

$$0 \leq \langle \boldsymbol{y}', \boldsymbol{x} \rangle \leq 2. \tag{D.2}$$

Now, we define a function $d_\Delta(v) : V \to \mathbb{R}$ by assigning $d_\Delta(v) := d_{\max}(v) - d_{\min}(v) \in [0, 2]$ for each vertex $v \in V$. In this case,

$$\mathcal{I}_V := \sum_{v \in V} 2\boldsymbol{x}[v] \ln \boldsymbol{x}[v] = \sum_{v \in V} d_\Delta(v) \cdot \boldsymbol{x}[v] \ln \boldsymbol{x}[v] + \sum_{v \in V} \left(2 - d_\Delta(v)\right) \cdot \boldsymbol{x}[v] \ln \boldsymbol{x}[v]$$

$$= \sum_{v \in V \backslash \{\mathsf{s}\}} d_\Delta(v) \sum_{e \in \delta^-(v)} \boldsymbol{x}[e] \ln \boldsymbol{x}[v] + \sum_{v \in V \backslash \{\mathsf{t}\}} \left(2 - d_\Delta(v)\right) \sum_{e \in \delta^+(v)} \boldsymbol{x}[e] \ln \boldsymbol{x}[v] \tag{D.3}$$

Furthermore, for every edge $(u, v) \in E$, we define

$$\boldsymbol{y}''[(u, v)] := d_\Delta(v) + \boldsymbol{y}'[(u, v)] - d_\Delta(u) = d_{\max}(v) + \boldsymbol{y}[(u, v)] - d_{\max}(u).$$

By the triangle inequality, we have $\boldsymbol{y}''[e] \leq 0$ for all $e \in E$. Similarly, from the properties of $\boldsymbol{y}'$ and $d_\Delta(\cdot)$, we also have $\boldsymbol{y}''[e] \geq -2$ for any edge $e \in E$. Furthermore, we can write

$$\mathcal{I}'_E := \sum_{e \in E} \left(2 + \boldsymbol{y}''[e]\right) \cdot \boldsymbol{x}[e] \ln \boldsymbol{x}[e]$$

$$= \sum_{(u, v) \in E} \left(d_\Delta(v) + \boldsymbol{y}'[(u, v)] + 2 - d_\Delta(u)\right) \cdot \boldsymbol{x}[(u, v)] \ln \boldsymbol{x}[(u, v)]$$

$$= \sum_{v \in V \backslash \{\mathsf{s}\}} d_\Delta(v) \sum_{e \in \delta^-(v)} \boldsymbol{x}[e] \ln \boldsymbol{x}[e] + \sum_{e \in E} \boldsymbol{y}'[e] \cdot \boldsymbol{x}[e] \ln \boldsymbol{x}[e] + \sum_{v \in V \backslash \{\mathsf{t}\}} \left(2 - d_\Delta(v)\right) \sum_{e \in \delta^+(v)} \boldsymbol{x}[e] \ln \boldsymbol{x}[e] \tag{D.4}$$

As a result, we have

$$\psi(\boldsymbol{x}) = \underbrace{\frac{1}{2} \sum_{e \in E} \left(2 + \boldsymbol{y}''[e]\right) \boldsymbol{x}[e] \ln \boldsymbol{x}[e] - \frac{1}{2} \sum_{e \in E} \boldsymbol{y}''[e] \cdot \boldsymbol{x}[e] \ln \boldsymbol{x}[e]}_{\mathcal{I}'_E} - \underbrace{\frac{1}{2} \sum_{v \in V} 2\boldsymbol{x}[v] \ln \boldsymbol{x}[v]}_{\mathcal{I}_V}$$

$$= \underbrace{\frac{1}{2} \sum_{v \in V \backslash \{\mathsf{s}\}} d_\Delta(v) \sum_{e \in \delta^-(v)} \boldsymbol{x}[e] \ln \left(\frac{\boldsymbol{x}[e]}{\boldsymbol{x}[v]}\right)}_{\mathcal{I}^+} + \underbrace{\frac{1}{2} \sum_{v \in V \backslash \{\mathsf{t}\}} \left(2 - d_\Delta(v)\right) \sum_{e \in \delta^+(v)} \boldsymbol{x}[e] \ln \left(\frac{\boldsymbol{x}[e]}{\boldsymbol{x}[v]}\right)}_{\mathcal{I}^-}$$

$$+ \underbrace{\frac{1}{2} \sum_{e \in E} \boldsymbol{y}'[e] \cdot \boldsymbol{x}[e] \ln \boldsymbol{x}[e]}_{\mathcal{I}'} + \underbrace{\frac{1}{2} \sum_{e \in E} -\boldsymbol{y}''[e] \cdot \boldsymbol{x}[e] \ln \boldsymbol{x}[e]}_{\mathcal{I}''}$$

where the second equality follows from (D.3) and (D.4). Since $d_\Delta(v) \in [0, 2]$ and $\boldsymbol{y}''[e] \leq 0$, we have that $\mathcal{I}^+, \mathcal{I}^-$, and $\mathcal{I}''$ are all sums of convex functions. Therefore, their sum is also convex. Thus, for any vector $\boldsymbol{z} \in \mathrm{span}(\mathcal{X})$,

$$\boldsymbol{z}^\top \nabla^2_{\boldsymbol{x}\boldsymbol{x}} \left(\mathcal{I}^+ + \mathcal{I}^- + \mathcal{I}''\right) \boldsymbol{z} \geq 0.$$

Furthermore, from the second-order derivative $(x \ln x)'' = 1/x$, we have

$$\boldsymbol{z}^\top \nabla^2_{\boldsymbol{x}\boldsymbol{x}} \mathcal{I}' \boldsymbol{z} = \sum_{e \in E} \boldsymbol{z}[e]^2 \cdot \frac{\boldsymbol{y}'[e]}{\boldsymbol{x}[e]} \geq \left(\sum_{e \in E} \boldsymbol{z}[e]^2 \cdot \frac{\boldsymbol{y}'[e]}{\boldsymbol{x}[e]}\right) \cdot \frac{1}{2} \left(\sum_{e \in E} \boldsymbol{x}[e] \cdot \boldsymbol{y}'[e]\right)$$

$$\geq \frac{1}{2} \left(\sum_{e \in E} \boldsymbol{z}[e] \cdot \sqrt{\frac{\boldsymbol{y}'[e]}{\boldsymbol{x}[e]}} \cdot \sqrt{\boldsymbol{x}[e] \cdot \boldsymbol{y}'[e]}\right)^2 = \frac{1}{2} \langle \boldsymbol{y}', \boldsymbol{z} \rangle^2.$$

The first inequality follows from $\langle \boldsymbol{x}, \boldsymbol{y}' \rangle \leq 2$ in (D.2), and the second inequality follows from the Cauchy–Schwarz inequality. Plugging in the two inequalities above gives

$$\|\boldsymbol{z}\|^2_{\nabla^2 \psi(\boldsymbol{x})} := \boldsymbol{z}^\top \nabla^2 \psi(\boldsymbol{x}) \boldsymbol{z} \geq \frac{1}{4} \langle \boldsymbol{y}', \boldsymbol{z} \rangle^2. \tag{D.5}$$

Furthermore, we can also decompose the regularizer as

$$\psi(\boldsymbol{x}) = \underbrace{\sum_{v \in V \setminus \{\mathsf{s},\mathsf{t}\}} \sum_{e \in \delta^+(v)} \boldsymbol{x}[e] \ln\left(\frac{\boldsymbol{x}[e]}{\boldsymbol{x}[v]}\right)}_{\mathcal{I}^E} + \underbrace{\sum_{e \in \delta^+(\mathsf{s})} \boldsymbol{x}[e] \ln \boldsymbol{x}[e]}_{\mathcal{I}^V}.$$

Since $\mathcal{I}^E$ is the sum of convex functions, for any $\boldsymbol{z} \in \mathrm{span}(\mathcal{X})$, we have $\boldsymbol{z}^\top \nabla^2_{\boldsymbol{x}\boldsymbol{x}} \mathcal{I}^E \boldsymbol{z} \geq 0$. From the second order derivative $(x \ln x)'' = 1/x$, we have that

$$\boldsymbol{z}^\top \nabla^2_{\boldsymbol{x}\boldsymbol{x}} \mathcal{I}^V \boldsymbol{z} = \sum_{e \in \delta^+(\mathsf{s})} \boldsymbol{z}[e]^2 \cdot \frac{1}{\boldsymbol{x}[e]} \geq \left(\sum_{e \in \delta^+(\mathsf{s})} \boldsymbol{z}[e]^2 \cdot \frac{1}{\boldsymbol{x}[e]}\right) \cdot \left(\sum_{e \in \delta^+(\mathsf{s})} \boldsymbol{x}[e]\right)$$

$$\geq \left(\sum_{e \in \delta^+(\mathsf{s})} \boldsymbol{z}[e] \cdot \sqrt{\frac{1}{\boldsymbol{x}[e]}} \cdot \sqrt{\boldsymbol{x}[e]}\right)^2$$

$$\geq \big(d_{\min}(\mathsf{s}) \cdot \boldsymbol{z}[\mathsf{s}]\big)^2$$

where the second inequality follows from the Cauchy-Schwarz inequality, and the last inequality follows from $\big|d_{\min}(\mathsf{s})\big| \leq 1$. This indicates

$$\|\boldsymbol{z}\|^2_{\nabla^2 \psi(\boldsymbol{x})} := \boldsymbol{z}^\top \nabla^2 \psi(\boldsymbol{x}) \boldsymbol{z} \geq \big(d_{\min}(\mathsf{s}) \cdot \boldsymbol{z}[\mathsf{s}]\big)^2 \tag{D.6}$$

Combining (D.5) and (D.6) concludes that

$$10\|\boldsymbol{z}\|^2_{\nabla^2 \psi(\boldsymbol{x})} = 10\boldsymbol{z}^\top \nabla^2 \psi(\boldsymbol{x}) \boldsymbol{z} \geq 2\big|\langle \boldsymbol{y}', \boldsymbol{z}\rangle\big|^2 + 2\big(d_{\min}(\mathsf{s}) \cdot \boldsymbol{z}[\mathsf{s}]\big)^2 \geq \big(\langle \boldsymbol{y}', \boldsymbol{z}\rangle + d_{\min}(\mathsf{s}) \cdot \boldsymbol{z}[\mathsf{s}]\big)^2 = \langle \boldsymbol{y}, \boldsymbol{z}\rangle^2$$

where the second inequality follows from $2a^2 + 2b^2 \geq (a+b)^2$ for any $a, b \in \mathbb{R}$, and the last equality follows from (D.1). This completes the proof.

$\square$

**Lemma 6.4.** The dilated entropy $\psi$ is differentiable and strictly convex on the relative interior $C := \mathrm{relint}(\mathrm{co}(\mathcal{X}))$. Moreover, $\lim_{n \to \infty} \|\nabla_{\boldsymbol{x}} \psi(\boldsymbol{x}_n)\|_2 = \infty$ if $\{\boldsymbol{x}_n\}_n$ is sequence of points in $C$ approaching the boundary of $C$.

*Proof.* Let $C := \mathrm{relint}(\mathrm{co}(\mathcal{X}))$ and let $\partial C := \mathrm{co}(\mathcal{X}) \setminus C$. As every coordinate is positive inside the relative interior $C$, $\psi$ is differentiable on $C$. Due to Lemma 6.3, it follows that $\psi$ is also strictly convex on $C$.

Next consider a sequence $\{\boldsymbol{x}_n\}_n$ with $\boldsymbol{x}_n \in C$ for all $n$ and $\lim_{n \to \infty} \boldsymbol{x}_n = \boldsymbol{x}^* \in \partial C$. Consider a vertex $u \in V$ such that $\boldsymbol{x}^*[u] > 0$ and there exists an edge $e = (u, v)$ such that $\boldsymbol{x}^*[e] = 0$. Note that such a vertex $u$ exists otherwise $\boldsymbol{x}^* \in C$. Now observe that

$$\lim_{n \to \infty} \big|\nabla_{\boldsymbol{x}} \psi(\boldsymbol{x}_n)[e]\big| = \lim_{n \to \infty} \big|\ln \boldsymbol{x}_n[e] - \ln \boldsymbol{x}_n[u]\big| = \infty.$$

Hence, we have $\lim_{n \to \infty} \|\nabla_{\boldsymbol{x}} \psi(\boldsymbol{x}_n)\|_2 = \infty$. $\square$

**Theorem 6.5.** OMD with dilated entropy is iterate-equivalent to HEDGE over the set of paths $\mathcal{X}$.

*Proof.* Due to Lemma 6.4, the solution of the following OMD equation lies in the relative interior of $\mathrm{co}(\mathcal{X})$:

$$\widetilde{\boldsymbol{x}}_t \leftarrow \underset{\boldsymbol{x} \in \mathrm{co}(\mathcal{X})}{\mathrm{argmin}} \left\{\eta \langle \boldsymbol{y}_{t-1}, \boldsymbol{x}\rangle + \mathcal{D}_\psi(\boldsymbol{x} \,\|\, \widetilde{\boldsymbol{x}}_{t-1})\right\}$$

where $\psi(\boldsymbol{x})$ is the dilated entropy and $\eta = \sqrt{\log |\mathcal{X}|/T}$. That is, for all edges $e$, we have $\widetilde{\boldsymbol{x}}_t[e] > 0$.

For any vertex $v$, denote by $\delta^-(v)$ and $\delta^+(v)$ the sets of incoming and outgoing edges from $v$, respectively. Recall that $\text{co}(\mathcal{X})$ is described by the following linear constraints:

$$g_v(\boldsymbol{x}) := \sum_{e \in \delta^-(v)} \boldsymbol{x}[e] - \sum_{e \in \delta^+(v)} \boldsymbol{x}[e] = 0 \quad \forall v \in V \setminus \{\mathsf{s}, \mathsf{t}\}$$

$$g_\mathsf{s}(\boldsymbol{x}) := 1 - \sum_{e \in \delta^+(\mathsf{s})} \boldsymbol{x}[e] = 0$$

$$g_\mathsf{t}(\boldsymbol{x}) := \sum_{e \in \delta^-(\mathsf{t})} \boldsymbol{x}[e] - 1 = 0$$

$$h_e(\boldsymbol{x}) := -\boldsymbol{x}[e] \le 0$$

Let the Lagrangian be defined as:

$$\mathcal{L}_t(\boldsymbol{x}, \boldsymbol{\lambda}, \boldsymbol{\mu}) := \eta \langle \boldsymbol{y}_{t-1}, \boldsymbol{x} \rangle + \mathcal{D}_\psi(\boldsymbol{x} \,\|\, \widetilde{\boldsymbol{x}}_{t-1}) + \sum_{v \in V} \boldsymbol{\lambda}[v] g_v(\boldsymbol{x}) + \sum_{e \in E} \boldsymbol{\mu}[e] h_e(\boldsymbol{x})$$

By the KKT conditions, there exists a tuple $(\widetilde{\boldsymbol{x}}_t, \boldsymbol{\lambda}_t, \boldsymbol{\mu}_t)$ such that

$$\left. \frac{\partial \mathcal{L}_t(\boldsymbol{x}, \boldsymbol{\lambda}, \boldsymbol{\mu})}{\partial \boldsymbol{x}[e]} \right|_{\boldsymbol{x} = \widetilde{\boldsymbol{x}}_t,\ \boldsymbol{\lambda} = \boldsymbol{\lambda}_t,\ \boldsymbol{\mu} = \boldsymbol{\mu}_t} = 0.$$

Due to complementary slackness and the fact that $\widetilde{\boldsymbol{x}}_t[e] > 0$ for all $e \in E$, it follows that $\boldsymbol{\mu}_t[e] = 0$ for all $e \in E$. Hence, we obtain:

$$\ln \frac{\widetilde{\boldsymbol{x}}_t[e]}{\widetilde{\boldsymbol{x}}_t[u]} = \ln \frac{\widetilde{\boldsymbol{x}}_{t-1}[e]}{\widetilde{\boldsymbol{x}}_{t-1}[u]} - (\eta \boldsymbol{y}_{t-1}[e] - \boldsymbol{\lambda}_t[u] + \boldsymbol{\lambda}_t[v])$$

where $e = (u, v)$.

Let $\mathbb{P}_t(p)$ denote the probability of choosing path $p$ in round $t$. Observe that

$$\mathbb{P}_t(p) = \prod_{e=(u,v) \in p} \frac{\widetilde{\boldsymbol{x}}_t[e]}{\widetilde{\boldsymbol{x}}_t[u]} \propto \prod_{e=(u,v) \in p} \frac{\widetilde{\boldsymbol{x}}_{t-1}[e]}{\widetilde{\boldsymbol{x}}_{t-1}[u]} \exp\left( -\eta \sum_{e=(u,v) \in p} \boldsymbol{y}_{t-1}[e] \right)$$

$$= \mathbb{P}_{t-1}(p) \exp\left( -\eta \sum_{e=(u,v) \in p} \boldsymbol{y}_{t-1}[e] \right).$$

Thus, OMD with dilated entropy is equivalent to HEDGE over paths. $\qquad \square$

# E  Another Near-optimal OMD for DAGs

Let $G = (V, E)$ be a DAG, and let $\mathcal{X} \subseteq \{0, 1\}^E$ represent all paths from the source vertex $\mathsf{s}$ to the sink vertex $\mathsf{t}$. According to Maiti et al. [32], there always exists a strategically equivalent DAG in which the longest $\mathsf{s}$-$\mathsf{t}$ path contains at most $\mathcal{O}(\log |\mathcal{X}|)$ edges. Therefore, without loss of generality, we assume that the longest path from $\mathsf{s}$ to $\mathsf{t}$ contains $\mathcal{O}(\log |\mathcal{X}|)$ edges.

Let $\boldsymbol{y}_t \in \mathbb{R}^E$ be the loss vector observed in round $t$. We construct a modified non-negative loss vector $\boldsymbol{y}_t' \in \mathbb{R}_{\ge 0}^E$ such that:

- There exists a constant $\alpha_t$ such that $\langle \boldsymbol{x}, \boldsymbol{y}_t' \rangle = \langle \boldsymbol{x}, \boldsymbol{y}_t \rangle + \alpha_t$ for any $\boldsymbol{x} \in \mathcal{X}$.
- For any $\boldsymbol{x} \in \mathcal{X}$, it satisfies $\sum_{e \in E} \boldsymbol{x}[e] \boldsymbol{y}_t'[e]^2 \le 4$.

Given the weight vector $\boldsymbol{y}_t$, we define, for each vertex $v \in V$, the quantities $d_{\min}(v) \in \mathbb{R}$ and $d_{\max}(v) \in \mathbb{R}$ as the weights of the shortest and longest paths from $\mathsf{s}$ to $v$, respectively. Clearly, $d_{\min}(\mathsf{s}) = d_{\max}(\mathsf{s}) = 0$, and $-1 \le d_{\min}(\mathsf{t}) \le d_{\max}(\mathsf{t}) \le 1$.

We define the modified loss vector $\boldsymbol{y}_t' \in \mathbb{R}^E$ by assigning:

$$\boldsymbol{y}_t'[(u, v)] = \boldsymbol{y}_t[(u, v)] + d_{\min}(u) - d_{\min}(v).$$

Then, for any $x \in \mathcal{X}$, we have $\langle x, y'_t \rangle = \langle x, y_t \rangle - d_{\min}(\mathsf{t})$.

We now prove that $\sum_{e \in E} x[e] y'_t[e]^2 \leq 4$ for any $x \in \mathcal{X}$. Define a function $d_\Delta : V \to \mathbb{R}$ by $d_\Delta(v) := d_{\max}(v) - d_{\min}(v) \in [0, 2]$ for each $v \in V$. Note that $y'_t[(u, v)] \geq 0$ by the triangle inequality. Furthermore, using the inequality $d_{\max}(v) \geq d_{\max}(u) + y_t[(u, v)]$, we get:

$$y'_t[(u, v)] \leq d_{\max}(v) - d_{\max}(u) + d_{\min}(u) - d_{\min}(v) = d_\Delta(v) - d_\Delta(u).$$

Now, consider any path $x \in \mathcal{X}$ with vertices $v_0 = \mathsf{s}, v_1, \ldots, v_k = \mathsf{t}$. We have:

$$\sum_{e \in E} x[e] y'_t[e]^2 = \sum_{i=1}^{k} y'_t[(v_{i-1}, v_i)]^2 \leq \sum_{i=1}^{k} \left( d_\Delta(v_i) - d_\Delta(v_{i-1}) \right)^2$$

$$\leq \left( \sum_{i=1}^{k} \left( d_\Delta(v_i) - d_\Delta(v_{i-1}) \right) \right)^2 = d_\Delta(\mathsf{t})^2 \leq 4.$$

Now consider the OMD update rule:

$$\widetilde{x}_t \leftarrow \operatorname*{argmin}_{\widetilde{x} \in \mathrm{co}(\mathcal{X})} \left\{ \eta \langle y'_{t-1}, \widetilde{x} \rangle + \mathcal{D}_\phi(\widetilde{x} \,\|\, \widetilde{x}_{t-1}) \right\}$$

where the regularizer is defined as

$$\phi(\widetilde{x}) := \sum_{e \in E} \left( \widetilde{x}[e] \ln \widetilde{x}[e] - \widetilde{x}[e] \right).$$

Let $\| \cdot \|_{\nabla^2 \phi(\widetilde{x}_t)}$ denote the local norm induced by the Hessian of $\phi$ at $\widetilde{x}_t$, and the corresponding dual norm is given by $\| \cdot \|_{\nabla^{-2} \phi(\widetilde{x}_t)}$. Using standard regret analysis, we have:

$$\mathbb{E}[\mathrm{Regret}(T)] \leq \frac{1}{\eta} \left( \max_{\widetilde{x} \in \mathrm{co}(\mathcal{X})} \phi(\widetilde{x}) - \min_{\widetilde{x} \in \mathrm{co}(\mathcal{X})} \phi(\widetilde{x}) \right) + \eta \sum_{t=1}^{T} \| y'_t \|^2_{\nabla^{-2} \phi(\widetilde{x}_t)}$$

$$\leq \frac{1}{\eta} \max_{\widetilde{x} \in \mathrm{co}(\mathcal{X})} \sum_{e \in E} \left( \widetilde{x}[e] \ln(1/\widetilde{x}[e]) + \widetilde{x}[e] \right) + \eta \sum_{t=1}^{T} \| y'_t \|^2_{\nabla^{-2} \phi(\widetilde{x}_t)}$$

$$\leq \frac{c}{\eta} \cdot \log |\mathcal{X}| \cdot \log d + \eta \sum_{t=1}^{T} \sum_{e \in E} \widetilde{x}_t[e] y'_t[e]^2$$

$$\leq \frac{c}{\eta} \cdot \log |\mathcal{X}| \cdot \log d + 4\eta \cdot T$$

where $c$ is an absolute constant. The third inequality uses the fact that the number of edges in any path is upper bounded by $\mathcal{O}(\log |\mathcal{X}|)$, along with Jensen's inequality. The fourth inequality follows from the construction of $y'_t$. By setting $\eta = \sqrt{T^{-1} \log |\mathcal{X}| \cdot \log d}$, the regret is upper bounded by $\mathcal{O}\left( \sqrt{T \log |\mathcal{X}| \cdot \log d} \right)$.

Having established an alternative OMD algorithm for DAGs, we now outline a general template for designing near-optimal OMD algorithms using the negative entropy regularizer for other combinatorial sets $\mathcal{X} \subseteq \{0, 1\}^d$.

- **Lift the action set**: Map $\mathcal{X}$ to an equivalent higher-dimensional set $\widetilde{\mathcal{X}} \subseteq \{0, 1\}^D$, where $D = \mathrm{poly}(d)$, such that the diameter of $\widetilde{\mathcal{X}}$ is small with respect to the negative entropy regularizer.

- **Change the loss vectors**: Construct an equivalent non-negative loss vector such that the sum of squared dual norms is small.

## F   Auxiliary Lemmas

**Lemma F.1** (Yao's lemma)**.** Let $\mathcal{A}$ be a set of deterministic algorithms, and $\mathcal{Y}$ a set of inputs. Let $c(A, y)$ denote the cost incurred by a deterministic algorithm $A \in \mathcal{A}$ on input $y \in \mathcal{Y}$, such as its regret.

Let $\mathcal{D}$ be a distribution over inputs and $\mathcal{R}$ be a class of distributions over deterministic algorithms. Then for any randomized algorithm $R \in \mathcal{R}$, we have:

$$\min_{A \in \mathcal{A}} \mathbb{E}_{y \sim \mathcal{D}}[c(A, y)] \leq \max_{y \in \mathcal{Y}} \mathbb{E}_R[c(R, y)].$$

**Remark:** By Yao's lemma, to prove a lower bound on the expected cost of any randomized algorithm, it suffices to exhibit a distribution over inputs under which every deterministic algorithm incurs high expected cost.

**Lemma F.2** (Khintchine inequality (restated), Haagerup [26])**.** Let $\{\xi_t\}_{t=1}^T$ be $T$ independent Rademacher random variables with $\mathbb{P}(\xi_t = \pm 1) = 1/2$. Let $x_1, \ldots, x_T \in \mathbb{C}$. Then,

$$\sqrt{\frac{1}{2} \sum_{t=1}^T |x_t|^2} \leq \mathbb{E}\left| \sum_{t=1}^T \xi_t x_t \right| \leq \sqrt{\sum_{t=1}^T |x_t|^2}.$$

**Lemma F.3** (Orabona and Pál [35])**.** There exists universal constants $c_1 \geq 8$ and $c_2 > 0$ such that for any $d \in [c_1, \exp(T/3)]$, we have

$$\mathbb{E}\left[ \max_{i \in [\![d]\!]} \sum_{t=1}^T \xi_{t,i} \right] \geq c_2 \cdot \sqrt{T \log d}.$$

where $\{\xi_{t,i}\}_{t=1, i=1}^{T, d}$ are i.i.d. Rademacher random variables with $\mathbb{P}(\xi_{t,i} = \pm 1) = 1/2$.

**Lemma F.4.** For any $2 \leq K \leq \exp(T/3)$, there exists a zero-mean distribution $\mathcal{D}_K$ over $\{-1, +1\}^K$ such that if $\boldsymbol{y}_t \sim \mathcal{D}_K$ for $t = 1, \ldots, T$, then

$$-\mathbb{E}\left[ \min_{i \in [\![K]\!]} \sum_{t=1}^T \langle \boldsymbol{e}_i, \boldsymbol{y}_t \rangle \right] \geq c_0 \cdot \sqrt{T \log K},$$

where $c_0 > 0$ is some universal constant

*Proof.* Consider the universal constants $c_1$ and $c_2$ from Lemma F.3. If $2 \leq K < c_1$, we define the distribution $\mathcal{D}_K$ as the uniform distribution over $\{\boldsymbol{e}_1, -\boldsymbol{e}_1\}$. If $\boldsymbol{y}_t \sim \mathcal{D}_K$ for $t = 1, \ldots, T$, then by applying the same analysis as in Theorem 3.2, we obtain:

$$-\mathbb{E}\left[ \min_{i \in [\![K]\!]} \sum_{t=1}^T \langle \boldsymbol{e}_i, \boldsymbol{y}_t \rangle \right] \geq \sqrt{T/8} \geq \sqrt{T \log K/(8 \log c_1)}.$$

On the other hand, if $K \geq c_1$, we define the distribution $\mathcal{D}_K$ as the uniform distribution over $\{-1, +1\}^K$. If $\boldsymbol{y}_t \sim \mathcal{D}_K$ for $t = 1, \ldots, T$, then by applying the Lemma F.3, we obtain:

$$-\mathbb{E}\left[ \min_{i \in [\![K]\!]} \sum_{t=1}^T \langle \boldsymbol{e}_i, \boldsymbol{y}_t \rangle \right] = \mathbb{E}\left[ \max_{i \in [\![K]\!]} \sum_{t=1}^T \langle \boldsymbol{e}_i, -\boldsymbol{y}_t \rangle \right] \geq c_2 \cdot \sqrt{T \log K}.$$

$\square$

# G    Numerical Lemmas

**Lemma G.1.** Let $a, b > 0$ be two numbers such that $a \geq b$. Then

$$\left\lfloor \frac{a}{b} \right\rfloor \geq \frac{a}{2b}.$$

*Proof.* If we set

$$x = \frac{a}{b} \geq 1, \quad n = \lfloor x \rfloor,$$

then

$$x = n + k, \quad 0 \leq k < 1,$$

so

$$\frac{a}{2b} = \frac{x}{2} = \frac{n+k}{2} \leq \frac{n+1}{2}.$$

Since $n \geq 1$,

$$\frac{n+1}{2} \leq n \iff n+1 \leq 2n \iff n \geq 1.$$

Hence

$$\frac{a}{2b} = \frac{x}{2} \leq n = \left\lfloor \frac{a}{b} \right\rfloor.$$

$\square$

**Lemma G.2.** For any $a > 3$, $(a/x)^x$ is an increasing in the range $[1, a/3]$.

*Proof.* It suffices to show that for every real $x \in [1, a/4]$ the derivative of

$$f(x) := \left(\frac{a}{x}\right)^x$$

is strictly positive. Consider

$$g(x) := \ln f(x) = x \ln \left(\frac{a}{x}\right) = x \ln a - x \ln x,$$

so

$$g'(x) = \ln a - (\ln x + 1) = \ln \left(\frac{a}{x}\right) - 1.$$

Whenever $x \leq a/3$, we have

$$\frac{a}{x} \geq \frac{a}{a/3} = 3 \quad \implies \quad \ln \left(\frac{a}{x}\right) \geq \ln 3.$$

Since $\ln 3 \approx 1.098 > 1$, it follows that

$$g'(x) = \ln \left(\frac{a}{x}\right) - 1 \geq \ln 3 - 1 > 0.$$

Because $f'(x) = f(x)g'(x)$ and $f(x) > 0$, we obtain

$$f'(x) > 0 \quad \text{for all } 1 \leq x \leq a/3.$$

$\square$

**Lemma G.3.** Denote by

$$\mathcal{X} := \left\{ \boldsymbol{x} \in \{0,1\}^d : \sum_{i=1}^{d} \boldsymbol{x}[i] = m \right\}.$$

$$\mathcal{S} := \left\{ \boldsymbol{x} \in \mathcal{X} : |\{i \in [\![m]\!] : \boldsymbol{x}[i] \neq 1\}| \geq \left\lfloor \frac{m}{20} \right\rfloor \right\}.$$

When $m \in [\![20, d/2]\!]$, it satisfies that

$$\frac{|\mathcal{X} \setminus \mathcal{S}|}{|\mathcal{X}|} \leq \exp\left( -\frac{m}{20} \cdot \ln \left(\frac{d}{m}\right) \right)$$

*Proof.* We upper bound $|\mathcal{X} \setminus \mathcal{S}|$ by enumerating on possible values of $k := \left|\{i \in [\![m]\!] : \boldsymbol{x}[i] \neq 1\}\right| = \left|\{i \notin [\![m]\!] : \boldsymbol{x}[i] = 1\}\right|$.

$$
\begin{aligned}
|\mathcal{X} \setminus \mathcal{S}| &= \sum_{k=0}^{\lfloor m/20 \rfloor - 1} \binom{m}{k} \binom{d-m}{k} \\
&\leq \frac{m}{20} \cdot \binom{m}{\lfloor m/20 \rfloor} \binom{d}{\lfloor m/20 \rfloor} \\
&\leq \frac{m}{20} \cdot \left(\frac{em}{\lfloor m/20 \rfloor}\right)^{\lfloor m/20 \rfloor} \left(\frac{ed}{\lfloor m/20 \rfloor}\right)^{\lfloor m/20 \rfloor} && \text{(as } \binom{d}{m} \leq \left(\frac{ed}{m}\right)^m \text{)} \\
&\leq \frac{m}{20} \cdot (20e)^{m/20} \left(\frac{20ed}{m}\right)^{m/20} && \text{(due to Lemma G.2)} \\
&\leq \frac{(m/20) \cdot (20e)^{m/10} (d/m)^{m/20}}{(d/m)^m} \cdot \binom{d}{m} && \text{(as } (d/m)^m \leq \binom{d}{m} \text{)} \\
&\leq \frac{(m/20) \cdot (20e)^{m/10}}{2^{9m/10}} \cdot (m/d)^{m/20} \cdot \binom{d}{m} && \text{(as } m \leq d/2 \text{)} \\
&\leq \exp\left(-\frac{m}{20} \cdot \ln\left(\frac{d}{m}\right)\right) \cdot \binom{d}{m} && \text{(as } \frac{(m/20) \cdot (20e)^{m/10}}{2^{9m/10}} \leq 1 \text{)}
\end{aligned}
$$

Since $|\mathcal{X}| = \binom{d}{m}$, we immediately reach the desired result. $\qquad\square$