# OpenReview forum: "On the Universal Near Optimality of Hedge in Combinatorial Settings"
_NeurIPS.cc/2025/Conference — NeurIPS 2025 spotlight_

### Official Review · Reviewer_aAm5 · 2025-06-30

**Clarity:** 3
**Significance:** 3
**Originality:** 2
**Rating:** 5
**Confidence:** 2

**Summary:**

The authors study the regret performance of HEDGE in combinatorial settings. They ask whether it’s universally optimal and conclude "nearly" by providing an algorithm-independent regret lower bound that match the well-known upper bound up to $\sqrt{\log d}$ factors.

They then provide some clarity on the nature of HEDGE's (sub-)optimality by studying two different combinatorial task instances where HEDGE is optimal and where it isn’t.

For the optimal case, they provide tighter algorithm-independent lower bounds for the online multiask learning setting, which match HEDGE's upper bound.

For the sub-optimal case, they show their original lower bound is tight for the $m$-sets task, and is enjoyed by a modified OMD algorithm. They also show that HEDGE incurs a strictly worse regret compared to the modified OMD by a $\sqrt{\log d}$ factor.

Finally, they show how regularized OMD is iterate-equivalent to HEDGE, and therefore enjoys near-optimal regret for shortest-path problems on DAGs.

**Questions:**

1. In the introduction, the adversary is described as potentially reactive, in that it can choose loss vectors as a function of the history $\mathcal{F}_t.$ However, in the main theorems, the adversary is described as oblivious, fixing the losses at the start of the trial. I may have misunderstood something, but can the authors please comment on whether they are focusing on an adaptive or oblivious adversary?

2. Intuitively, why is the structure of the $m$-sets task poorly aligned with HEDGE while online multitask learning is well aligned? Currently the paper requires the reader to engage with the full proof to appreciate the differences. A discussion that helps the reader more readily appreciate the essential differences would be a valuable contribution to this work.

**Ethical Concerns:**

["NO or VERY MINOR ethics concerns only"]

**Final Justification:**

This paper provides compelling theoretical contributions to help clarify the nature of HEDGE’s (sub-)optimality on an important class of online learning problems. The organisation is very clear, the focus of the paper builds well, and natural questions are addressed. The authors addressed my only questions sufficiently, and I stand by my original rating of accept.

**Limitations:**

The authors should add some additional commentary addressing the weaknesses I mentioned.

**Paper Formatting Concerns:**

No major formatting issues.

**Quality:**

3

**Strengths And Weaknesses:**

*Strengths:*
- Compelling theoretical contributions to help clarify the nature of HEDGE’s (sub-)optimality on an important class of online learning problems
- Organisation is very clear
- Focus of the paper builds well, natural questions are addressed
- FIgure nicely summarizes contributions

*Weaknesses:*
- While a gap of $\sqrt{\log d}$ is non-trivial, it may represent a very modest gap for many of the cited applications. The authors should add a short section discussing the significance of this gap in context of their cited motivations.
- Although the main theorems are well stated and clear, the paper lacks a discussion on the essence of what makes $m$-sets a particularly challenging task in comparison to online multitask learning for HEDGE. The authors should add a short section providing some intuition for why one might expect HEDGE to be optimal in one but not in the other.
- The paper mentions an adaptive adversary in the preliminaries (lines 139-140), but in the provided theorems the adversary is oblivious. The authors should address this inconsistency.

---

> ### Author Rebuttal · Authors · 2025-07-31
>
> We thank the reviewer for the positive feedback. We now address their concerns below.
>
> **Reviewer's question:**
> Can the authors please comment on whether they are focusing on an adaptive or oblivious adversary?
>
> **Author's response:**
> The upper bound results for our algorithms hold against any type of adversary, including adaptive ones. For the lower bound, it suffices to consider an oblivious adversary, since a lower bound in this setting also implies a lower bound in the adaptive case.
>
> ---
>
> **Reviewer's question:**
> Intuitively, why is the structure of the m-sets task poorly aligned with HEDGE while online multitask learning is well aligned? [...] A discussion that helps the reader more readily appreciate the essential differences would be a valuable contribution to this work.
>
> **Author's response:**
> We appreciate the reviewer’s suggestion. At a high level, HEDGE operates without exploiting the underlying structure of the decision set and treats each element of $X$ as an individual expert. This aligns well with settings like online multitask learning, where the decision set is a Cartesian product of simplices (HEDGE is known to be optimal for simplices), and this structure allows HEDGE to perform optimally.
>
> In contrast, m-sets impose a strong global constraint: each vector must contain exactly $m$ ones. This introduces dependencies among coordinates that HEDGE fails to leverage. As a result, HEDGE incurs regret scaling with $\log |X|$, whereas more structure-aware algorithms (such as OMD with a tailored regularizer) can achieve strictly better bounds. We agree that adding this discussion would improve the paper and will incorporate a brief section to highlight this intuition.
>
> Nonetheless, while the present work focuses on establishing regret bounds, a deeper understanding of the conditions under which HEDGE performs well or poorly could inform the development of more effective algorithms. We view this as a promising direction for future work.
>
> ---
>
> **Reviewer's comment:**
> While a gap of $\log d$ is non-trivial, it may represent a very modest gap for many of the cited applications. The authors should add a short section discussing the significance of this gap in context of their cited motivations.
>
> **Author's response:**
> We thank the reviewer for the suggestion and will incorporate this change in the final version.

---

> > ### Comment · Reviewer_aAm5 · 2025-08-05
> >
> > Thank you for your response and for addressing my questions. I will keep my score.
> >
> > Note: upon re-reading with your rebuttal in mind I noticed a small typo: "there exists a" is repeated twice in Theorem 4.6.

---

### Official Review · Reviewer_fnGB · 2025-07-02

**Clarity:** 4
**Significance:** 3
**Originality:** 4
**Rating:** 5
**Confidence:** 4

**Summary:**

The paper studies the classical Hedge algorithm for online learning in combinatorial settings. The main contribution is a series of upper and lower bounds on the regret that establish near-optimality of Hedge. In particular, the paper identifies subproblems of the combinatorial setting where Hedge is provable optimal respectively suboptimal by a factor $\log(d)$.

**Questions:**

Are you aware of setting where Hedge achieves the minimax lower bound $\sqrt{ T \log|X|/\log(d)}$?

**Ethical Concerns:**

["NO or VERY MINOR ethics concerns only"]

**Final Justification:**

I would like to thank the authors for the detailed response. After reading the reviews and the author replies, my overall evaluation stays positive and I recommend acceptance.

**Limitations:**

While the paper addresses the regret bounds for the Hedge algorithm (and minimax bounds), one would hope to have a general characterisation of the regret for any given X, e.g. by introducing a complexity measure that depends on the set X (either combinatorial or even in the general case). Another open question is to design an algorithm that is near-optimal for a larger family or all combinatorial sets.

**Quality:**

4

**Strengths And Weaknesses:**

This paper makes an important contribution to the online learning literature. Hedge is a classical algorithms but there are still open questions around its optimality in various settings. This paper focuses on the combinatorial setting, where the decision set is a subset of $\\{0,1\\}^d$. The authors first establish that any algorithm suffers a worst-case regret of $\Omega(\sqrt{T \log(|X|)/ \log d})$ ,where X is the decision set. The known upper bound of Hedge is $O(\sqrt{T \log(|X|)}$, hence the question is if the gap is loose or if one of the bounds can be improved. The paper subsequently establishes further lower and upper bounds for the Hedge algorithm, showing cases where Hedge is provably suboptimal, and settings where the lower bound can be improved.

The paper is very well written and easy to follow. Related work is discussed appropriately.

Some minor suggestions for improvement are as follows:
* consider include definitions of important quantities, such es worst-case and minimax regret, and iterate-equivalence.
* In line 194, I assume you invoke an existing regret lower bound over the set I. Consider citing or stating the result again in this context.
* Line 198: What is the expectation over here? How does the expected regret related to the original objective defined in line 142?

---

> ### Author Rebuttal · Authors · 2025-07-31
>
> We thank the reviewer for the positive feedback. We now address their concerns below.
>
> **Reviewer's question:**
> Are you aware of a setting where Hedge achieves the minimax lower bound $\sqrt{T \log |X| / \log d}$?
>
> **Author's response:**
> We are not aware of a setting where Hedge achieves the minimax lower bound $\sqrt{T \log |X| / \log d}$. We conjecture that Hedge has a lower bound of $\sqrt{T \log |X|}$ for any combinatorial set $X$.
>
> ---
>
> **Reviewer's comment:**
> Consider including definitions of important quantities, such as worst-case and minimax regret, and iterate-equivalence.
>
> **Author's response:**
> We thank the reviewer for the suggestion and will incorporate this change in the final version.
>
> ---
>
> **Reviewer's comment:**
> In line 194, I assume you invoke an existing regret lower bound over the set $\mathcal{I}$. Consider citing or stating the result again in this context.
>
> **Author's response:**
> Our result is in the appendix. We will state the result again in the main body.
>
> ---
>
> **Reviewer's comment:**
> Line 198: What is the expectation over here? How does the expected regret relate to the original objective defined in line 142?
>
> **Author's response:**
> The algorithm might be randomized and hence the expectation is over the randomness of the algorithm.
>
> ---
>
> **Reviewer's comment (limitations):**
> One would hope to have a general characterisation of the regret for any given $X$, e.g., by introducing a complexity measure that depends on the set $X$.
>
> **Author's response:**
> We agree that a general characterization of regret in terms of a complexity measure tailored to the decision set $X$, as well as the design of algorithms that are near-optimal for broader families of non-combinatorial sets, are important directions for future work. In particular, identifying variants of HEDGE that achieve near-optimal regret for arbitrary sets $X$ remains a compelling and open research question. Our work takes a step in this direction by establishing the near-optimality of HEDGE up to a $\sqrt{\log d}$ factor for all $X \subseteq ${$0,1$}$^d$, and showing regimes where this gap is provably unavoidable, but we fully agree that closing this gap in general remains an open challenge.

---

> > ### Comment · Reviewer_fnGB · 2025-08-05
> >
> > I would like to thank the authors for the detailed response. After reading the reviews and the author replies, my overall evaluation stays positive and I recommend acceptance.

---

### Official Review · Reviewer_v7MV · 2025-07-02

**Clarity:** 3
**Significance:** 2
**Originality:** 3
**Rating:** 5
**Confidence:** 3

**Summary:**

This paper studies the problem of prediction with expert advice in a combinatorial setting (i.e., full-information setting). While most of previous works focus on characterizing regret's upper bounds of various algorithms, this paper focus on constructing hard examples for establishing regret's lower bounds and from there, analyze the optimality (or sub-optimality) of the famous HEDGE algorithm. In particular, the authors show that Hedge is near-optimality (up to a logarithm term of the dimension) in any setting, is strictly suboptimal in a special class, and optimal in another class of problem.

**Questions:**

- Line 776 (Appendix A) assume that $T$ divides $|\mathcal{I}|$. If this is not the case, how does Inequality in Line 787 works? In particulatr, if we partition the T rounds into $|\mathcal{I}| - 1$ equal-length segments and 1 last segment that is smaller than that. Does this mean that there is a set {t : it = i} whose size is strictly smaller than $T$ / |\mathcal{I}|$. As a consequence Inequality in Line 787 is a " strictly less than" rather than a "less or equal" ?

- Is Lemma F.3 a restatement of Theorem 7 of (Orabona and Pál [35])? If that's the case, can you explain where does the condition c1 >= 8 comes from (or which term is it equivalent to in Orabona andPal [35])?
Moreover, while this is a bit picky, if it is possible, I think you should follow the  template of Orabona and Pal  to obtain the explicit value of the involved constant. The reason is because such constant in Orabona and Pal contains the term log(d) in it; therefore, it is better to check and make sure that we do not misjudge the order of log(d) when writing big-oh(sqrt(Tlog(d)).
This is quite important as this constant in Lemma F.3 leads to constant in Lemma F.4 and hence, directly involves in Theorem 5.1.

**Ethical Concerns:**

["NO or VERY MINOR ethics concerns only"]

**Final Justification:**

The authors have provided adequate responses to my questions. The only remaining concern is the lack of a numerical experiment showing a hard-case where standard algorithms fall into the theoretically proven bound.

After reading the other reviewers' opinions, I opt to maintain my original score.

**Limitations:**

Yes

**Paper Formatting Concerns:**

No major formatting issues

**Quality:**

3

**Strengths And Weaknesses:**

Strengths:
- The paper is quite clearly written and sufficiently easy to follow.
- As far as I know, the contribution is clear and original.
- Significance: While the key question is quite simple, it is interesting to establish the lower bound guarantees of HEDGE as it is a well-used algorithm.

Weaknesses:
- All proofs are put in Appendices so it is a bit hard to have an immediate intuition where does the result come from.
- The notation  Phi $\varphi$ is used and reused repeatedly throughout various sections to refer to different choices/variants of the regularizer. It would be better if the authors distinguish these notations; for example, denote Phi_{m-set} in (4.1) and  Phi_{entropy} in 6.1.
- A minor point: all results contain implicit constants that might involve a leading term (for example, dimension d, see questions for more details).
The paper limits to the full-information setting. It would be interesting to make connection to previously known results of regret's lower bounds in other setting (for example, bandit).

The implicit constants make it hard to check the proof.

Lacks some numerical experiments or concrete example to confirm the results. For instance, construct a toy example of a hard-case and show the numerical regret of several algorithms would be interesting.

---

> ### Author Rebuttal · Authors · 2025-07-31
>
> We thank the reviewer for the positive feedback. We now address their concerns below.
>
> **Reviewer's comment:**
> All proofs are put in Appendices so it is a bit hard to have an immediate intuition where does the result come from.
> The notation $\Phi$ is used and reused repeatedly throughout various sections to refer to different choices/variants of the regularizer.
>
> **Author's response:**
> We will provide more intuition and change the notation accordingly in the final version.
>
> ---
>
> **Reviewer's question:**
> Line 776 (Appendix A) assume that $T$ divides $|\mathcal{I}|$. If this is not the case, how does Inequality in Line 787 work?
>
> **Author's response:**
> The analysis for the case when $|\mathcal{I}|$ does not divide $T$ proceeds identically to the case where $|\mathcal{I}|$ divides $T$. The only difference is that for the first $T' = |\mathcal{I}| \cdot \lfloor T/|\mathcal{I}| \rfloor$ rounds, we use the same hard instance, and for the remaining rounds $T'+1, T'+2, \ldots, T$, we assign loss vectors that are zero across all coordinates. Since $T' \geq T/2$, this still yields the lower bound $\Omega(\sqrt{T \log |X| / \log d})$.
>
> ---
>
> **Reviewer's comment and question:** All results contain implicit constants that might involve a leading term (for example, dimension d). Is Lemma F.3 a restatement of Theorem 7 of (Orabona and Pál [35])? If that's the case, Where does the condition $c_1 \geq 8$ come from (or which term is it equivalent to in Orabona and Pál \[35])?
>
> **Author's response:**
> Yes, Theorem F.3 in our paper is a restatement of Theorem 7. The expectation is lower bounded by $0.09\sqrt{n \ln d} - 2\sqrt{n}$, which in turn is at least $0.01\sqrt{n \ln d}$ whenever $\sqrt{\ln d} \geq 25$. Hence, there exists a universal constant $c_1$ (clearly greater than 8 which is arbitrarily chosen) and independent of $d$, such that Theorem F.3 holds for all $d \in [c_1,\exp(T/3)]$. As a result, none of the bounds and constants in the paper implicitly hide any dependence on $d$.

---

### Official Review · Reviewer_vJEE · 2025-07-03

**Clarity:** 3
**Significance:** 3
**Originality:** 2
**Rating:** 5
**Confidence:** 2

**Summary:**

This paper studies online learning over combinatorial decision sets $ \\mathcal{X} \subset \\{0,1\\}^d $,
where at each round the learner selects an action $x_t \in \mathcal{X}$, observes the loss vector $y_t$, and incurs loss$x_t^\bot y_t$. The goal is to minimize cumulative regret with respect to the best fixed action in hindsight. The authors revisit the performance of the classical Hedge algorithm in the combinatorial setting. They show a general lower bound proving that Hedge is at most order $\sqrt{\log d}$ worst-case sub-optimal compared to its best known upper bounds.  The authors  demonstrate that this lower bound is tight for certain combinatorial structures; specifically, for m-sets where this bound is attained by an instance of mirror descent;  while Hedge  is provably suboptimal,  as proven by a lower bound on its regret for this combinatorial structure.  Finally, the authors show that Hedge is minimax optimal in other combinatorial structures such as online multitask learning and shortest-path problems in DAG.

**Questions:**

Does Theorem 4.6 still hold if Hedge is run with a decreasing learning rate? In particular, could an adaptive rate improve the result, or does the lower bound remain unaffected?

**Ethical Concerns:**

["NO or VERY MINOR ethics concerns only"]

**Final Justification:**

The authors have successfully addressed my minor concerns during rebuttal.  And the results presented here could benefit the community in understanding the strengths and weaknesses of the well-adopted Hedge algorithm. I recommend acceptance.

**Limitations:**

yes

**Paper Formatting Concerns:**

No formatting concerns

**Quality:**

3

**Strengths And Weaknesses:**

**Strengths**: The problem studied is particularly relevant in applications, given the widespread use and simplicity of Hedge in practice.
The authors establish a worst-case lower bound for general combinatorial structures, showing that the Hedge algorithm is minimax optimal up to a$\sqrt{⁡\log d}$ factor. They demonstrate the tightness of this bound for m-sets, where the optimal regret is achieved by an instance of OMD, while also showing that it is provably not attainable by Hedge for some loss sequences. The authors show matching upper and lower bounds for Hedge on DAGs and in combinatorial structure for multitask learning. These results offers valuable insight into the complexity combinatorial online learning.

**Weaknesses**:
- the lower bounds presented are worst-case in nature, which makes it difficult to determine whether the (sub)optimality arises inherently from the combinatorial structure itself or from carefully constructed adversarial instances (that could belong to other strucutures as well). While this is understandable given the adversarial setting, it limits the interpretability of the results across broader classes of structures.
- the results are somewhat ad hoc and tailored to each specific combinatorial domain, making it nontrivial to identify a unifying complexity measure that governs the complexity of combinatorial learning

**Minor comments**:
- typo in l.232
- $co(X)$ is used l.145 before being introduced l.182

---

> ### Author Rebuttal · Authors · 2025-07-31
>
> We thank the reviewer for the positive feedback. We now address their concerns below.
>
> **Reviewer's comment:**
> The lower bounds presented are worst-case in nature [...] The results are somewhat ad hoc and tailored to each specific combinatorial domain, making it nontrivial to identify a unifying complexity measure that governs the complexity of combinatorial learning.
>
> **Author's response:**
> We appreciate the reviewer’s concern. While our lower bounds are worst-case, they do not stem from carefully constructed adversarial instances but rather arise inherently from the combinatorial structure itself. This is evident in the proof of our main general result (Theorem 3.2), where we establish that for any set $X \subseteq ${$0,1$}$^d$, the regret is lower bounded by
>
> $$
> \Omega\left(\sqrt{T \cdot \frac{\log |X|}{\log d}}\right),
> $$
>
> via a general reduction based on the Sauer–Shelah lemma. Combined with the $O(\sqrt{T \log |X|})$ regret bound of HEDGE, this shows that both the upper and lower bounds scale with $\log |X|$, suggesting that it serves as a natural and interpretable measure of complexity.
>
> The additional lower bounds we provide for specific domains such as m-sets and multitask learning are not tailored constructions, but are intended to demonstrate that the general bound is tight in some cases and loose in others—specifically up to a $\sqrt{\log d}$ factor. These examples clarify when HEDGE is optimal and when it is provably suboptimal.
>
> ---
>
> **Reviewer's question:**
> Does Theorem 4.6 still hold if Hedge is run with a decreasing learning rate?
>
> **Author's response:**
> We thank the reviewer for raising this question. Decreasing learning rates are typically used to obtain anytime guarantees, while adaptive learning rates (e.g., AdaHedge) aim to improve regret on easier instances. However, these modifications are not known to improve the worst-case regret of HEDGE. Although Theorem 4.6 analyzes HEDGE with a fixed learning rate, we believe the lower bound should continue to hold, possibly under a modified construction, even for variants with decreasing or adaptive learning rates. Formalizing this remains an interesting direction for future work.

---

> > ### Comment · Reviewer_vJEE · 2025-08-06
> >
> > I would like to thank the authors for their  thoughtful response to my comments. I appreciate their efforts to clarify the technical aspects and to provide further intuition behind the results. I maintain my positive evaluation of this work and recommend it for acceptance.

---

### Official Review · Reviewer_3jBY · 2025-07-07

**Clarity:** 4
**Significance:** 4
**Originality:** 3
**Rating:** 6
**Confidence:** 4

**Summary:**

It is well known that the Hedge algorithm guarantees an external regret of O(sqrt{T log d}) in the learning with experts setting, and that this bound is tight up to constant factors. It is possible to apply Hedge to more general online learning settings -- for example, for online linear optimization when the action set is the polytope, one can run an instance of Hedge where each expert corresponds to a vertex of the polytope, getting an O(sqrt{T log V}) bound. But is this regret bound tight?

This paper studies this question for the case of *combinatorial action sets*, where the action set P of the learner is the convex hull of a subset of points in {0, 1}^d (named as it captures many combinatorial sets of interest). The authors prove the following set of results:

1. First, they show that the regret bound of Hedge is within a sqrt(log d) factor of optimal -- in particular, that for any combinatorial polytope, there is a regret lower bound of Omega(sqrt{T log V / log d}). The proof of this result is a consequence of the Sauer-Shelah theorem (which bounds the number of distinct 0/1 points needed to contain a subset that projects to a hypercube).
2. Second, they demonstrate cases where this separation is tight; specifically, they show that this separation is tight for m-sets (sets of points with Hamming weight m), by constructing a new optimal regularizer for these points (a convex combination of the quadratic and negative entropy regularizer).
3. Thirdly, they show that Hedge is minimax optimal for all “Online Multitask Learning” problems (essentially, where the action set of the learner is the product of several simplices).
4. Finally, they also show that Hedge is minimax optimal when P is the flow polytope in a DAG. They also show that Hedge can be interpreted as running FTRL with the dilated entropy regularizer.

**Questions:**

See above -- in general, feel free to reply to any aspect of the above review.

**Ethical Concerns:**

["NO or VERY MINOR ethics concerns only"]

**Final Justification:**

After reading the other reviews and the author's response, my (positive) score remains unchanged. I think this is a very strong NeurIPS submission (making non-trivial progress on a very natural theoretical learning question), and was my favorite of submissions I reviewed this year.

**Limitations:**

None.

**Quality:**

3

**Strengths And Weaknesses:**

I think this is a very cool set of results! Surprisingly little is known about what the optimal minimax regret algorithms are for instance of online linear optimization -- while it is well understood in very classical settings (e.g. the learner plays actions in the l_1 or l_2 norm ball and the adversary plays actions in the polar), as soon as the learner / adversary action set becomes a little bit more complex it is not clear what the optimal regret bounds or algorithms are for. This is despite this being a very fundamental primitive in many applications (e.g. game theoretic applications, online learning in combinatorial domains). This paper provides a fairly thorough answer to this question for a very important class of action sets (aside from cases like the l_2 ball, it is hard to think of natural settings in the literature this does not capture).

One possible criticism of this paper is that the proofs are fairly simple (they are generally fairly immediate consequences of the correct observations). But I think despite this, these observations were certainly not well known to other practioners in the field (I have spent some time thinking about these questions, and none of these facts were obvious to me), and so I think there is a sizable subset of the NeurIPS audience that would appreciate these observations.

Question: Section 6 only addresses the unit flow polytope formed by the convex hull of all single paths. Do you know if this result continues to hold for general flow polytopes (either allowing for any positive flow or a specific positive flow > 1?)? My guess is that the answer is no (and that the m-set example can be embedded in the higher flow polytope of some DAG -- e.g., maybe one where the max flow is m but there are d > m edges of capacity 1 connecting two vertices), but I haven’t thought carefully about this.

---

> ### Author Rebuttal · Authors · 2025-07-31
>
> We thank the reviewer for the positive feedback. We now address their concerns below.
>
> **Reviewer's question:**
> Section 6 only addresses the unit flow polytope formed by the convex hull of all single paths. Do you know if this result continues to hold for general flow polytopes (either allowing for any positive flow or a specific positive flow > 1?)?
>
> **Author's response:**
> It remains an open question whether this result extends to general flow polytopes. We have given some thought to how one might design variants of Hedge to handle this general case. Resolving this case could yield tighter regret bounds for non-combinatorial sets.

---

### Decision · Program_Chairs · 2025-09-17

**Decision:**

Accept (spotlight)

**Comment:**

This paper provides worst-case near-optimal bounds for Hedge in several combinatorial settings. The reviewing team was overwhelmingly positive about the submission. Clear accept. Good job!

Please, incorporate all the feedback received from the reviewers in the revised version.